# Uncertainty and Risk Evaluation of Deep Geothermal Energy Source for Heat Production and Electricity Generation in Remote Northern Regions

**Mafalda M. Miranda [1],\*** , **Jasmin Raymond [1]** and **Chrystel Dezayes [2]**

1   INRS—Institut National de la Recherche Scientifique, 490 Rue de la Couronne,
    Québec, QC G1K 9A9, Canada; jasmin.raymond@inrs.ca
2   BRGM, F-45060 Orléans, France; c.dezayes@brgm.fr
\*   Correspondence: mafalda_alexandra.miranda@ete.inrs.ca

**Abstract:** The Canadian off-grid communities heavily rely on fossil fuels. This unsustainable energetic framework needs to change, and deep geothermal energy can play an important role. However, limited data availability is one of the challenges to face when evaluating such resources in remote areas. Thus, a first-order assessment of the geothermal energy source is, therefore, needed to trigger interest for further development in northern communities. This is the scope of the present work. Shallow subsurface data and outcrop samples treated as subsurface analogs were used to infer the deep geothermal potential beneath the community of Kuujjuaq (Nunavik, Canada). 2D heat conduction models with time-varying upper boundary condition reproducing climate events were used to simulate the subsurface temperature distribution. The available thermal energy was inferred with the volume method. Monte Carlo-based sensitivity analyses were carried out to determine the main geological and technical uncertainties on the deep geothermal potential and risk analysis to forecast future energy production. The results obtained, although speculative, suggest that the old Canadian Shield beneath Kuujjuaq host potential to fulfill the community's annual average heating demand of 37 GWh. Hence, deep geothermal energy can be a promising solution to support the energy transition of remote northern communities.

**Keywords:** geothermal energy; geothermal gradient; paleoclimate; numerical model; Monte Carlo method; heat-in-place; theoretical potential; technical potential; petrothermal system; Nunavik

## 1. Introduction

The access to electricity is still a global challenge nowadays. In 2016, there were about 1 billion people off-grid worldwide [1]. In Canada, 239 communities rely solely on fossil fuels for electricity, space heating, and domestic hot water [2,3]. In most of these communities, the fuel is shipped only once a year with long-term storage period in sometimes old facilities having a risk of spill [3]. Moreover, the electricity cost is volatile and more than 50% higher than in southern Canada [3,4]. Therefore, the volatility and high cost of the diesel price, the will for higher energy security, and the severe environmental consequences brought interest to develop local, sustainable, and carbon-free energy resources, not only in the Canadian remote communities but also in other off-grid areas worldwide (e.g., [1,5–19]). Geothermal energy can be one of such renewable options to replace diesel consumption and provide electricity and heating/cooling for both Arctic/subarctic (e.g., [20–24]) and non-Arctic (e.g., [25–28]) remote and off-grid regions. Compared with other sources of renewable energy, geothermal has a high capacity factor and is available indefinitely regardless of weather conditions [29]. Ground-coupled heat pumps are believed to be an interesting heating

alternative for the residential dwellings [30,31] and to support greenhouses food production [32] in such arctic and subarctic climate. Borehole thermal energy storage has also been studied and can be a promising technology to help improve energy and food security in the arctic/subarctic environment [33]. Furthermore, geothermal systems of all kinds can be integrated with other renewable sources and technologies, enhancing their individual efficiency in cold climates [34]. Technological advances in the geothermal energy sector now allow to envision the exploitation of deep geothermal energy source in geological environments other than hydrothermal systems. For example, the engineered geothermal systems (EGS) concept (e.g., [35]), together with binary cycle geothermal power plants (GPP), can generate electricity from low-temperature resources (e.g., [36]).

Unfortunately, an important data gap exists in northern territories to accurately assess the local and deep geothermal energy source potential (cf. [2,37,38]). For this reason, efforts have been made to adapt methodologies and draw guidelines using outcrops as subsurface analogs to provide initial data for preliminary geothermal energy source assessment associated with petrothermal systems. Thus, an evaluation of the geothermal energy source and potential heat and power output is presented in this work to forecast future energy production. It is convenient to highlight that the term geothermal energy source was utilized in this work to follow the United Nations Framework Classification on fossil energy and mineral reserves and resources 2009 to geothermal energy resources [39].

Guidelines to carry out geothermal energy source and potential power output assessment associated with petrothermal systems have been proposed in the literature (e.g., [35,40,41]) and were adopted in this work to account for heat production and cogeneration. These previous studies differ in terminology and somewhat calculation methods. Tester et al. [35] followed the resources terminology of Muffler and Cataldi [42], while Beardsmore et al. [40] and Limberger et al. [41] followed the geothermal potential classification of Rybach [43]. Moreover, Limberger et al. [41] extended Beardsmore et al. [40] protocol and evaluated the levelized cost of energy and the economic potential. Regardless of the protocol followed, the basic element of the geothermal energy source and potential heat and power output assessment is the estimation of the available thermal energy, or "heat-in-place". The volume method introduced by USGS (United States Geological Survey) researchers (e.g., [42,44,45]) is the most widely used evaluation technique to infer the available thermal energy (e.g., [35,37,40,41,46–51]). This method is based on the evaluation of the heat stored in a certain volume of rock at specified depths in relation to the mean annual surface temperature [42]. However, many authors argued that, for a more realistic assessment, the reference temperature should be equivalent to the reservoir abandonment temperature. This is dependent on the intended application (space heating and/or electricity generation) and on the type of GPP to be installed (e.g., [40,49,52]). The second key element is the recoverable fraction. This concept was introduced since only a fraction of the available thermal energy can be harvested. This is mostly due to technical and economic constraints, such as drilling depth and cost, the active stimulated volume, the allowed reservoir thermal drawdown, and the surface land area available [35,40,47]. Finally, the conversion of thermal energy to power output takes into account the project lifetime, the availability of the GPP throughout the year, and the cycle thermal efficiency [35,40,49].

An evaluation of the subsurface temperature distribution is imperative to calculate the available thermal energy accordingly with the volume method. Thus, 2D transient heat conduction models were solved numerically with finite element method (FEM). Several climate episodes have occurred throughout Earth's history (e.g., [53]) that propagates downwards by thermal diffusion influencing the subsurface temperature (e.g., [54,55]) and these shall not be ignored. Additionally, the effect of temperature and pressure on thermal conductivity was considered and implemented in the numerical models used to simulate the subsurface temperature. The variability of the rock thermal conductivity has been shown to influence these predictions (e.g., [56–58]).

Then, the assessment of the available thermal energy was constrained by the envisioned applications: heat production and electricity generation. The minimum temperature for space heating is about 30–50 °C [49,59] and for electricity generation using a binary cycle GPP designed for an

Arctic climate is about 120–140 °C [60]. Thus, these were used as reservoir abandonment temperatures. Nevertheless, Organic Rankine Cycle with an optimized working fluid may generate electricity from geothermal energy source lower than 120 °C (e.g., [61–64]). The thermal energy was assessed every 1 km in depth for a total depth of 10 km, for the land surface area occupied by the community of Kuujjuaq (ca. 4 km$^2$). Finally, a range of theoretical recovery factors was investigated. The planar fracture method developed by Bodvarsson [65] and Bodvarsson and Tsang [66], later modified by Williams [67], has been widely used to predict theoretical recovery factors values for fracture-dominated systems (8 to 20%; [68]). Sanyal and Butler [69], however, simulated recovery factors of about 40% for a stimulated rock volume higher than 0.1 km$^3$. In this work, a conservative range between 2% and 20% was preferred following the recommendations of Tester et al. [35,47] and Beardsmore et al. [40].

Afterward, thermal energy was converted to heat and power output. For the latter, a cycle net thermal efficiency correlation equation has been proposed by Tester et al. [35] and was used in this study. This equation estimates quantitatively the percentage of heat that can be converted to electricity by binary GPP, thus allowing for a first-order evaluation of the potential power output. The project lifetime is constrained by several economic and technical factors, such as the minimum economic limit, design life, reservoir sustainable management, maintenance, contract, and entitlement periods [39,49]. Often, a 30-year life cycle is assumed for the evaluation of the geothermal potential [35,40]. However, technical aspects may dictate a longer or shorter lifetime, and this needs to be considered to evaluate the potential heat and power output and plan future energy production. Therefore, project lifetimes of 20 to 50 years were examined in this study. The GPP factor usually varies between 90% and 97% [49] and this range was considered in this study.

Lastly, global sensitivity analysis with Monte Carlo simulations was undertaken in this study for uncertainty and risk evaluation [68,70]. The global sampling and probabilistic approach were preferred to account for the current geological and technical uncertainties considered in this study. Detailed explanations of the Monte Carlo method and global sensitivity analysis can be found in, for example, Rubinstein and Kroese [71], Graham and Talay [72], Thomopoulos [73], and Scheidt et al. [74]. Broadly, global sensitivity analysis based on Monte Carlo simulates the possible scenarios by random sampling the input variables jointly, within the defined span of the probability distribution functions. The use of these methods enabled to infer the most influential uncertainties and assess the probability of the deep geothermal energy source to meet the community's heat and power demand.

The influence of the statistical distribution of the bedrock thermophysical properties cannot be neglected [75] when doing such geothermal energy source assessment and this uncertainty was considered throughout this study. Moreover, the effect of water saturation on the thermophysical properties was considered as well. Although the water saturation cannot be readily observed at depth, its effect may lead to significant miscalculations and shall not be ignored (e.g., [76]).

Thus, this study is the first of its kind undertaken in the Canadian Shield and represents an initial step to assess if deep geothermal energy source can be a viable alternative for remote northern communities settled in that physiographic region. Nevertheless, the approach followed in this study can be extended to other remote areas facing the same off-grid challenges (e.g., Svalbard, Faeroe Islands, Greenland, and other Arctic and non-Arctic communities). The thermal energy and potential output for heat production and electricity and cogeneration were examined and the main current geological (both epistemic and aleatory variability) and technical uncertainties were determined by the sensitivity analysis carried out. The statistical distribution of the thermophysical properties due to their intrinsic heterogeneous character is an aleatory variability type. The subsurface temperature, the conditions of the thermophysical properties (dry and water saturation), and the climate signal during a glacial period are epistemic uncertainties that can be decreased with further geothermal exploration development. Reservoir abandonment temperature, recovery factor, project lifetime, and GPP factor are technical uncertainties that can be optimized to maximize the energy production. The outcomes of this first-order

assessment are useful to plan further geothermal developments and forecast future energy production, hence, helping remote northern communities to move toward a more sustainable energetic framework.

## 2. Geographic and Geological Setting

Nunavik is home to 14 communities that are independent of the southern provincial electrical grid and rely exclusively on diesel for electricity, space heating, and domestic hot water, like the majority of communities in northern Canada [2,3]. In this region, the price of fuel oil per liter amounted to $2.03 in 2018, which was subsidized to $1.63 by the local government [4]. Therefore, geothermal energy sources may be a solution for this unfavorable energetic framework. However, in Nunavik, a territory of about 507,000 km$^2$, only three deep boreholes exist to evaluate heat flux. Those are Raglan mine, Asbestos Hill mine, and camp Coulon, which are located away from the Inuit communities (Figure 1; [38] and references therein). This highlights the need of adapting geothermal exploration methodologies to use outcrops treated as subsurface analogs to obtain a first estimate of the geothermal potential.

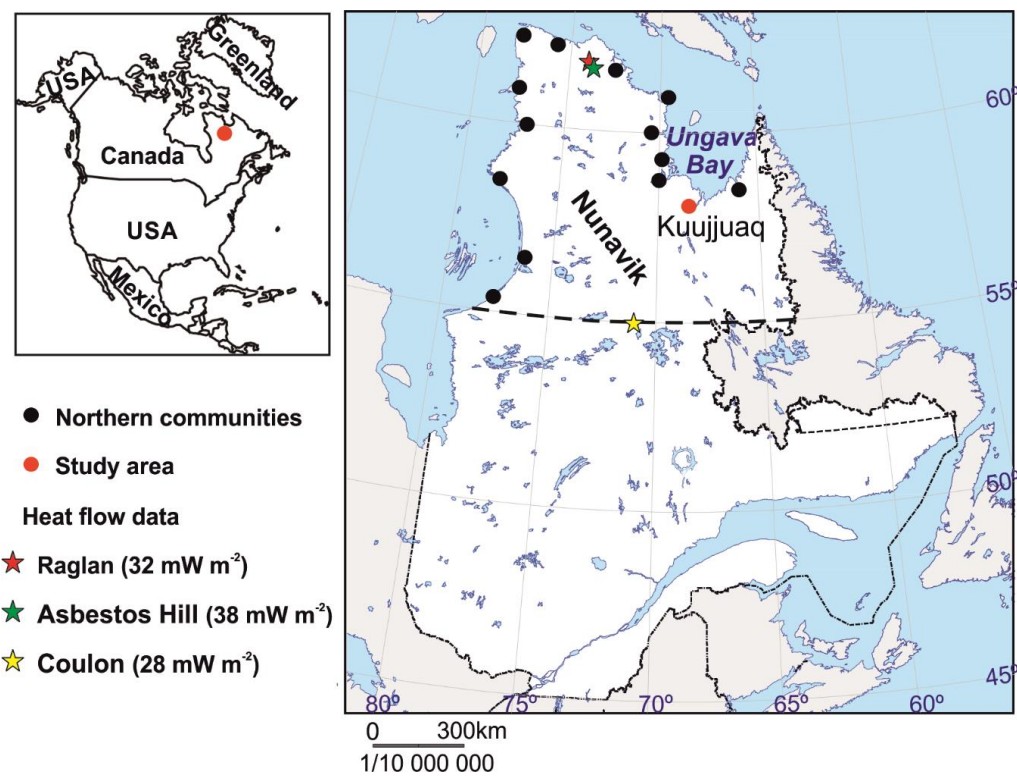

**Figure 1.** Geographical location of Kuujjuaq and the remaining Nunavik communities (Canada) along the shore of a vast territory with few heat flow assessments: camp Coulon, Raglan, and Asbestos Hill mining sites [38] and references therein (adapted from Miranda et al. [75]).

Nunavik is a vast territory, with the communities dispersed along the shore (Figure 1), and for that reason, a community-focused assessment was preferred to foster local deep geothermal development. Thus, avoiding to (1) extrapolate sparse data over such a large region and (2) blind the local potential by regional anomalies far away from the communities and useless for their energy transition. Kuujjuaq (Figure 1) is used as a case study and an example for the remaining communities. This village is the administrative capital of Nunavik and the largest within that territory, enclosing about 2750 inhabitants and 518 private dwellings [77] in a surface land area of approximately 4 km$^2$ [78]. In 2000, the annual electricity demand in Kuujjuaq was about 12,000 MWh [79], increasing to 15,100 MWh in 2011 [80]. The daily electricity consumption amounts typically to 15 to 22 kWh per dwelling, depending on the season, and is used exclusively for lighting and electrical household appliances [81]. Gunawan et al. [31] simulated the heating load of a typical 5-occupants residential dwelling in Kuujjuaq.

Their results reveal an annual heating energy demand of about 71 MWh per residence. These values provide a gross, first-order perspective of Kuujjuaq heat and power demand.

The main lithologic units outcropping near Kuujjuaq are paragneiss and diorite that were sampled in the framework of this study (Figure 2). These rocks belong to the Canadian Shield and are Neoarchean to Paleoproterozoic in age [82]. A detailed description of these units and the samples collected can be found in Miranda et al. [75] and references therein. The two main structures present in the study area are Lac Pingiajjulik and Lac Gabriel faults (Figure 2). Both are described as regional thrust faults with dextral movement [82].

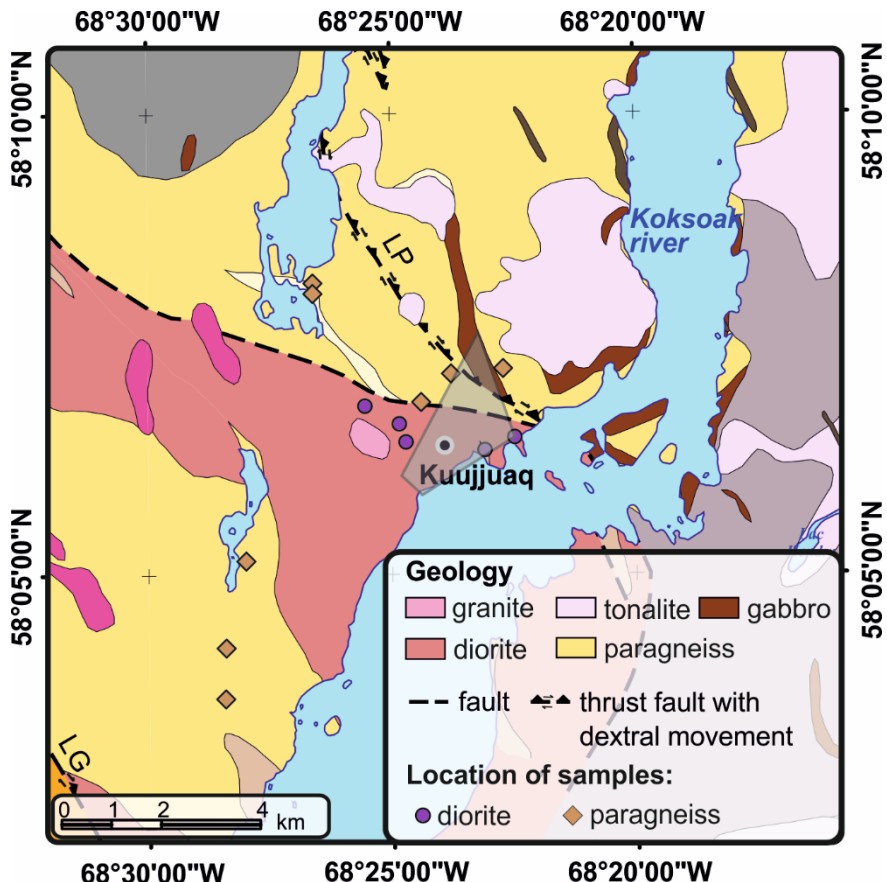

**Figure 2.** Geological map of the study area (adapted from SIGÉOM [82] and Miranda et al. [75]). LP—Lac Pingiajjulik fault, LG—Lac Gabriel fault. The grey polygon represents the surface land area occupied by Kuujjuaq's community [78].

## 3. Materials and Methods

### 3.1. Thermophysical Properties

A total of 13 rock samples were collected in the study area (Figure 2) and prepared for the laboratory analyses. Core plugs with 20-mm-radius and 20 to 30-mm-thickness were drilled from the hand samples. Then, the core plugs were analyzed for thermal conductivity and volumetric heat capacity at dry conditions with a guarded heat flow meter. In a second time, the plugs were placed in a vacuum chamber and immersed in water for 24 h to reach the water saturation state. Thermal conductivity and volumetric heat capacity were re-evaluated considering water-saturation. Porosity was additionally evaluated as a function of pressure to indirectly infer the effect of pressure on thermal conductivity. The concentration in uranium (U), thorium (Th), and potassium (K) was determined by gamma-ray spectrometry and inductively coupled plasma—mass spectrometry (ICP-MS).

Thermal conductivity and volumetric heat capacity were evaluated at both dry and water-saturated conditions in the laboratory using a FOX50 device from TA Instruments that has an accuracy of 3%. The device consists of two plates, two heat flow meters, and two insulating casings to prevent heat losses. The method follows the ASTM (American Society for Testing and Materials) standard C1784-13 (2013). The sample is placed between the plates and the temperature is allowed to reach equilibrium. A temperature difference of 10 °C is imposed on each plate for thermal conductivity assessment. The temperature of the plates is changed instantaneously for volumetric heat capacity evaluation and the time to reach equilibrium is needed to evaluate this property based on the energy conservation equation [83]. For both properties, successive data acquisition cycles grouped in blocks are run until all the necessary equilibrium criteria are reached and the sample is considered in thermal equilibrium (see Miranda et al. [75] for further details). Thermal conductivity was additionally evaluated within the temperature range of 20–160 °C to define an experimental relationship that describes the effect of temperature on thermal conductivity (e.g., [84]):

$$\frac{1}{\lambda(T)} = \frac{1}{\lambda_{20}} + b(T - 20) \Leftrightarrow \lambda(T) = \frac{1}{(\lambda_{20})^{-1} + b(T - 20)} \tag{1}$$

where $\lambda$ (W m$^{-1}$ K$^{-1}$) is thermal conductivity, $T$ (°C) is temperature, and $b$ is an experimental coefficient that controls temperature dependence of the thermal conductivity. The subscript 20 stands for room temperature. The effect of pressure on thermal conductivity was assessed indirectly from the pressure dependence on porosity. The combined gas permeameter-porosimeter AP-608 was used to evaluate porosity at different confining pressures from 2.8 to 69 MPa. The evaluation of porosity follows Boyle's law, which states that the pressure exerted by a given mass of an ideal gas is inversely proportional to the volume it occupies (e.g., [85] and references therein).

The results from this analysis were used to indirectly infer the effect of pressure on thermal conductivity, which is described by the following function:

$$\lambda(P) = d \ln(P) + \lambda_{20} \tag{2}$$

where $P$ (Pa) is pressure and $d$ is an experimental coefficient that controls the pressure dependence of thermal conductivity. The following relationship was then obtained when combining Equation (2) with Equation (1) to describe the effect of both temperature and confining pressure on thermal conductivity:

$$\lambda(T) = \frac{1}{(d \ln(P) + \lambda_{20})^{-1} + b(T - 20)} \tag{3}$$

The concentration in U, Th, and K was evaluated by both gamma-ray spectrometry and ICP-MS to avoid biased results (cf. [75]). The Ortec gamma-ray spectrometer detector used for this purpose is NaI(Tl) with 7.62 × 7.62 cm, surrounded by a 5-cm-thick lead shield. The concentrations of the radioisotopes were measured using the three-window method taking into account the emitted gamma radiation. The system is calibrated with standard solutions certified by the International Atomic Energy Agency (IAEA). The ICP-MS method, where the chemical elements passed through decomposition into their atomic constituents, was also used. The positively charged ions are extracted and separated, being finally measured by an ion detector. A quality control protocol was followed and certified reference materials used to guarantee the reliability of the analyses (see Miranda et al. [75] for further details).

Radiogenic heat production was then calculated by applying Rybach's empirical function [86]:

$$A = 10^{-5} \rho (9.51 C_{\text{U}} + 2.56 C_{\text{Th}} + 3.50 C_{\text{K}}) \tag{4}$$

where $A$ (W m$^{-3}$) is the radiogenic heat production, $\rho$ (kg m$^{-3}$) is the density, and $C$ (mg kg$^{-1}$; %) is the concentration of each radioisotope. The subscripts U, Th, and K stand for uranium, thorium, and potassium, respectively.

### 3.2. 2D Subsurface Temperature Distribution

The temperature-at-depth was solved numerically by FEM in COMSOL Multiphysics$^{®}$ with the 2D transient heat conduction equation:

$$\frac{\partial}{\partial x}\left(\lambda \frac{\partial T}{\partial x}\right) + \frac{\partial}{\partial z}\left(\lambda \frac{\partial T}{\partial z}\right) + A = \rho c \frac{\partial T}{\partial t} \tag{5}$$

where $x$ (m) and $z$ (m) are spatial variables, $\rho c$ (J m$^{-3}$ K$^{-1}$) is the volumetric heat capacity, and $t$ (s) is time. The geometry of the model is rectangular with width of 32 km and depth of 10 km (Figure 3) and takes into account the regional geological cross-section of Simard et al. [87].

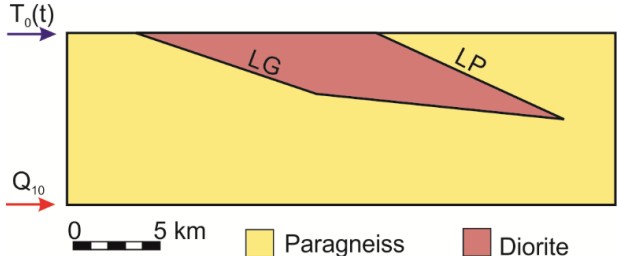

**Figure 3.** Simplified geological cross-section and model's geometry (based on Simard et al. [87]). LP—Lac Pingiajjulik fault, LG—Lac Gabriel fault; $T_0(t)$—time-varying upper boundary condition, $Q_{10}$—basal heat flux.

The initial temperature condition to run the transient simulations was calculated using the 1D analytical solution of Equation (5) in steady-state:

$$T(z) = T_0 + \frac{Q}{\lambda}z - \frac{A}{2\lambda}z^2 \tag{6}$$

The lateral boundary conditions were assumed adiabatic. The lower boundary condition is the basal heat flux. A numerical approach has been developed by Marquez et al. [88] and improved by Miranda et al. [89] to simulate climate events and find the basal heat flux that best matches a measured temperature log. The results of the latter were used in this work. The temperature profile was measured in an 80-m-deep groundwater monitoring well drilled prior to any of these studies. Considering the thermophysical properties at dry conditions, Miranda et al. [89] estimated the basal heat flux to range between 31.8 to 52.3 mW m$^{-2}$, with an average value of 41.6 mW m$^{-2}$. In turn, the thermophysical properties evaluated at water saturation conditions led to an increase of the heat flux: 34.1 to 69.4 mW m$^{-2}$, with an average value of 49.7 mW m$^{-2}$ [89]. The heat flux estimates of Miranda et al. [89] were used as the lower boundary condition in the present work.

A time-varying upper boundary condition was imposed to represent the ground surface temperature history (GSTH). COMSOL Multiphysics$^{®}$ piecewise function was used to implement the climate events (Table 1). The fourfold stratigraphic framework proposed by Flint [90] and Emiliani [91] to characterize the late Pleistocene (300–11.6 ka before present (B.P.)) climate events were considered in this study. The temperature at the base of the Laurentide Ice Sheet is still debatable. GSTH inversion from deep temperature profiles in Canada points toward an average temperature of −5 °C during the last glacial maximum, though lower temperatures were simulated in eastern Canada than in central Canada [92–96]. During the last glacial maximum, the Laurentide Ice Sheet reached a maximum thickness higher than 3 km [97], highlighting that under such thick ice the temperature was close to the

freezing point (−1 to −2 °C; [54,98]). A sensitivity study was undertaken to deal with the uncertainty of the Laurentide Ice Sheet basal temperature and its influence on the subsurface temperature (Table 1).

**Table 1.** Time and temperature steps of the major Pleistocene and Holocene climate events considered for ground surface temperature history (GSTH) in this work.

| Event | Time Step (Years B.P.) | Temperature Step (°C) | |
|---|---|---|---|
| | | Cold Scenario | Warm Scenario |
| Nebraskan (MIS 14 *) | 300,000–265,000 | −10 | −1 |
| Aftonian (MIS 13–11 *) | 265,000–200,000 | 0 | 0 |
| Kansan (MIS 10 *) | 200,000–175,000 | −10 | −1 |
| Yarmouth (MIS 9–7 *) | 175,000–125,000 | 0 | 0 |
| Illinoian (MIS 6 *) | 125,000–100,000 | −10 | −1 |
| Sangamonian (MIS 5 *) | 100,000–75,000 | 0 | 0 |
| Wisconsinan (MIS 4–2 *) | 75,000–11,600 | −10 | −1 |
| Holocene (MIS 1) | 11,600–present | | |
| Holocene thermal maximum | 7000–5800 | +2 | |
| Roman and Medieval warm periods | 3200–1000 | +1 | |
| Little Ice Age | 500–270 | −1 | |
| Pre-industrial and Industrial Revolution | 270–80 | +1.4 | |
| Present-day global warming | 30–present | +2 | |

* Based on Emiliani [91], MIS—Marine Isotope Stage, B.P.—before present.

The interglacial Holocene thermal maximum occurred ca. 7 – 5.8 ka B.P. ([99–101] and references therein) and is referred to have been 1–2 °C warmer than the present-day temperature [99,100,102,103]. The temperature during the interstadial Roman and Medieval warm periods (ca. 3.2–1 ka B.P.; [101] and references therein) were estimated to have been 1–1.5 °C warmer than at present [102]. During the stadial Little Ice Age (ca. 500–270 years B.P.; [101] and references therein), the temperature is estimated to have been 1 °C lower than today [102]. Majorowicz et al. [104] and Chouinard et al. [105] identified warming of about 1.4–2 °C during the pre-industrial and Industrial Revolution (ca. 270–80 years B.P.). This episode was followed by a short cooling episode (80–30 years B.P.), where temperature decreased around 0.4 °C [105]. Nowadays, meteorological data [106] can be converted empirically to undisturbed ground temperature with [107]:

$$T_g = 17.898 + 0.951 T_{amb} \tag{7}$$

where the subscripts g and amb are used for ground and ambient (air), respectively. This was done with Kuujjuaq historic weather data and revealed a sharp increase in the temperature of about 2 °C for the last 30 years.

The thermal properties of the geological materials were assumed at both dry and water-saturated state and the effect of temperature and pressure on thermal conductivity (Equation (3)) was implemented in the model. The statistical distribution of the thermophysical properties was also taken into account. The transient simulations were carried out for 300 ka with yearly time step to ensure a smooth solution for the effect of the more recent and short episodes of surface temperature changes. The backward differentiation formula was chosen for the time step method (e.g., [108]). The steps taken by the solver were set as free after a step-independency study have been undertaken.

*3.3. Geothermal Energy Source and Potential Heat and Power Output*

3.3.1. Volume Method

The available thermal energy content was assessed within the limits of Kuujjuaq land surface area covering 4 km² (Figure 2) down to 10 km. Volumetric heat capacity at both dry and water-saturated state was used in the calculations and its statistical distribution was considered as well. The available thermal energy was inferred with (e.g., [42]):

$$H = V\rho c(T_{res} - T_{ref})R \tag{8}$$

where $H$ (J) is the thermal energy, $V$ (m$^3$) is the volume, and $R$ (%) is the recovery factor. The subscripts res and ref stand for reservoir and reference temperature, respectively. The former was obtained through the 2D temperature simulations previously described and for the latter, the following hypotheses were assumed:

1. The reservoir abandonment temperature for space heating is about 30–50 °C [49,59]
2. The minimum temperature to generate electricity by a binary GPP considering an Arctic design is around 120–140 °C [60]

The recovery factor is not yet well constrained at this early stage of the geothermal exploration and, therefore, a theoretical range of 2–20% was used [35,40,47]. The conversion of the thermal energy to potential heat and power output (*PO*; W$_{th, e}$) was calculated with [35,40,49]:

$$PO = \frac{H\eta_{th}}{F_{GPP}t} \tag{9}$$

where $\eta$ (%) is the cycle efficiency and $F_{GPP}$ (%) is the GPP factor related with its availability throughout the year. The subscript th stands for thermal. The cycle net thermal efficiency was calculated as indicated by Tester et al. [35]:

$$\eta_{th} = 0.0935T_{res} - 2.3266 \tag{10}$$

The cycle thermal efficiency was only used to estimate the theoretical potential for electricity generation. The heat production evaluation did not consider this parameter since the heat energy is used directly (e.g., [109]). A range between 20 to 50 years of project lifetime was assumed. The GPP factor was varied between 90% and 97% [49]. It is important to highlight that no temperature loss was considered in this study [40].

### 3.3.2. Global Sensitivity Analysis with Monte Carlo Method

A global sensitivity analysis was undertaken to assess the joint effect of each parameter (and respective uncertainty; Table 2) on the potential heat and power output based on Monte Carlo method [74]. The simulations were carried out with @Risk [110] using Latin Hypercube sampling [111] and the pseudorandom number generator Marsenne Twister [112]. The Latin Hypercube sampling was chosen since it is referred to be more reliable and efficient than Monte Carlo sampling [113]. A total of 10,000 iterations (i.e., possible scenarios) were run per simulation to assure output stability. Moreover, the initial random number seed was fixed to 1 in all the simulations carried out. A total of 3 simulations without changing any of the inputs were run to confirm the solidity of the randomness of the sampling [113]. This approach was followed after carrying out an analysis of the stochasticity component of the response [74]. Five simulations were run and the difference in the output was less than 10%, indicating that the spatial uncertainty of the input parameters will have a minor impact on the deep geothermal energy source and potential heat and power output, and therefore can be neglected.

The existent GSTH and conditions of the thermophysical properties (dry or water-saturated) are unknown at this early stage of the geothermal development. Therefore, three hypotheses for the reservoir temperature were analyzed separately:

1. Thermophysical properties at dry conditions and warm GSTH
2. Thermophysical properties at dry conditions and cold GSTH
3. Thermophysical properties at water saturation conditions and warm GSTH

The outcomes from the uncertainty analysis can be translated to risk, enabling to forecast the probability of the deep geothermal energy source to meet the community's heat and power demand [114].

**Table 2.** Monte Carlo method input parameters and their uncertainty.

| Parameter Code | Parameter Description | Variable Type | Distribution |
|:---:|:---:|:---:|:---:|
| | Geological uncertainties | | |
| $V$ | Reservoir volume | | Single value |
| $T_{res}$ | Reservoir temperature | Continuous | Triang(min,median,max) |
| $\rho c$ | Volumetric heat capacity | Continuous | Normal($\mu$,$\sigma$) |
| | Technical uncertainties | | |
| $T_{ref}$ | Reservoir abandonment temperature | Continuous | Uniform(min,max) |
| $R$ | Recovery factor | Continuous | Uniform(min,max) |
| $\eta_{th}$ | Cycle thermal efficiency | | $f(T)$ |
| $F_{GPP}$ | GPP factor | Continuous | Uniform(min,max) |
| $t$ | Project lifetime | Continuous | Triang(min,most,max) |

Triang—Triangular probability distribution, min and max—minimum and maximum values, respectively, $\mu$—arithmetic mean, $\sigma$—population standard deviation, $f$—function, most—most likely value.

## 4. Results

### 4.1. Thermophysical Properties

At dry conditions and room temperature, the paragneiss samples are characterized by lower thermal conductivity than the diorite samples. The former has an average value of 2.26 W m$^{-1}$ K$^{-1}$, while the latter is characterized by an average thermal conductivity of 2.78 W m$^{-1}$ K$^{-1}$ (Table 3). Per contra, the volumetric heat capacity is higher for the paragneiss samples than for the diorite (Table 3). An average value of 2.32 MJ m$^{-3}$ K$^{-1}$ was inferred for the diorite while a value of 2.36 MJ m$^{-3}$ K$^{-1}$ was evaluated for the paragneiss. At water saturation conditions, the same trend is observed. The paragneiss samples have lower thermal conductivity but higher volumetric heat capacity than the diorite (Table 3). Likewise, higher concentration of radiogenic elements (U, Th, K) was evaluated for the paragneiss samples than for the diorite. This consequently influenced the inferred internal heat generation (Table 3). An average value of 1.08 µW m$^{-3}$ was inferred for the paragneiss while an average value of 0.53 µW m$^{-3}$ was evaluated for the diorite. These are average values from the two methods used in this work to evaluate the radiogenic element concentrations (gamma-ray spectrometry and ICP-MS).

**Table 3.** Results of the thermophysical properties analyses.

| | Paragneiss | | Diorite | |
|:---:|:---:|:---:|:---:|:---:|
| | **Dry** | **Wet** | **Dry** | **Wet** |
| | $\lambda$ (W m$^{-1}$ K$^{-1}$) | | | |
| $\mu$ | 2.26 | 2.67 | 2.78 | 3.08 |
| $\alpha$ | 0.55 | 0.64 | 0.65 | 0.82 |
| $\chi$ | 2.10 | 2.84 | 2.60 | 2.82 |
| [min–max] | 1.62–3.15 | 1.95–3.95 | 2.12–3.98 | 2.08–4.54 |
| | $\rho c$ (MJ m$^{-3}$ K$^{-1}$) | | | |
| $\mu$ | 2.36 | 2.44 | 2.32 | 2.36 |
| $\alpha$ | 0.10 | 0.18 | 0.14 | 0.12 |
| $\chi$ | 2.37 | 2.34 | 2.31 | 2.33 |
| [min–max] | 2.20–2.47 | 2.27–2.71 | 2.16–2.53 | 2.22–2.59 |
| | $A$ (µW m$^{-3}$) | | | |
| $\mu$ | 1.08 | | 0.53 | |
| $\alpha$ | 0.59 | | 0.41 | |
| $\chi$ | 1.16 | | 0.44 | |
| [min–max] | 0.21–1.99 | | 0.16–1.14 | |

$\lambda$—thermal conductivity, $\rho c$—volumetric heat capacity, $A$—radiogenic heat production, $\mu$—arithmetic mean, $\sigma$—population standard deviation, $\chi$—median, min and max—minimum and maximum values, respectively.

The thermal conductivity analysis of both paragneiss and diorite samples evaluated at dry conditions within the temperature range of 20 to 160 °C reveal a decrease between 18% to 40% as a function of temperature for the paragneiss samples (Table 4), while for the diorite samples the decrease is 34% to 52% (Table 5). The effect of pressure on thermal conductivity indirectly inferred (Equation (2)) reveal an increase of 3% to 15% for the paragneiss samples (Table 4) and 2% to 5% for the diorite samples (Table 5). The experimental coefficient *b* (Equation (1)) is found to range between 0.0003 and 0.002 for the paragneiss samples and between 0.0011 and 0.0051 for the diorite. The coefficient *d* (Equation (2)) varies within 0.02 and 0.20 and between 0.02 and 0.09 for the paragneiss and diorite samples, respectively.

**Table 4.** Thermal conductivity of the paragneiss samples as a function of temperature and pressure.

| | **Paragneiss** | | | | | | | |
|---|---|---|---|---|---|---|---|---|
| | $\lambda$**(W m$^{-1}$ K$^{-1}$)** | | | | | | | |
| *T* (°C) | 20 | 40 | 60 | 80 | 100 | 120 | 140 | 160 |
| $\mu$ | 2.32 | 2.28 | 2.22 | 2.11 | 2.05 | 1.95 | 1.88 | 1.63 |
| $\alpha$ | 0.63 | 0.63 | 0.66 | 0.66 | 0.76 | 0.72 | 0.71 | 0.63 |
| $\chi$ | 2.20 | 2.12 | 2.00 | 1.84 | 1.70 | 1.59 | 1.49 | 1.32 |
| [min–max] | 1.69–3.21 | 1.67–3.20 | 1.66–3.22 | 1.58–3.13 | 1.49–3.30 | 1.40–3.13 | 1.33–3.05 | 1.10–2.63 |
| | $\lambda$**(W m$^{-1}$ K$^{-1}$)** | | | | | | | |
| *P* (MPa) | 2.8 | 4.8 | 6.2 | 10.3 | 20.7 | 34.5 | 48.3 | |
| $\mu$ | 2.32 | 2.32 | 2.33 | 2.34 | 2.40 | 2.43 | 2.44 | |
| $\chi$ | 2.15 | 2.16 | 2.17 | 2.19 | 2.22 | 2.25 | 2.26 | |
| [min–max] | 1.64–3.32 | 1.64–3.43 | 1.65–3.53 | 1.66–3.67 | 1.67–3.79 | 1.70–3.86 | 1.68–3.92 | |

$\lambda$—thermal conductivity, *T*—temperature, *P*—pressure, $\mu$—arithmetic mean, $\sigma$—population standard deviation, $\chi$—median, min and max—minimum and maximum values, respectively.

**Table 5.** Thermal conductivity of the diorite samples as a function of temperature and pressure.

| | **Paragneiss** | | | | | | | |
|---|---|---|---|---|---|---|---|---|
| | $\lambda$**(W m$^{-1}$ K$^{-1}$)** | | | | | | | |
| *T* (°C) | 20 | 40 | 60 | 80 | 100 | 120 | 140 | 160 |
| $\mu$ | 2.39 | 2.33 | 2.25 | 2.10 | 1.96 | 1.86 | 1.78 | 1.49 |
| $\alpha$ | 0.87 | 0.83 | 0.81 | 0.78 | 0.80 | 0.77 | 0.76 | 0.68 |
| $\chi$ | 2.58 | 2.57 | 2.53 | 2.36 | 2.24 | 2.12 | 2.02 | 1.70 |
| [min–max] | 1.41–3.73 | 1.39–3.60 | 1.30–3.42 | 1.18–3.16 | 1.01–2.91 | 0.94–2.75 | 0.87–2.67 | 0.68–2.33 |
| | $\lambda$**(W m$^{-1}$ K$^{-1}$)** | | | | | | | |
| *P* (MPa) | 2.8 | 4.8 | 6.2 | 10.3 | 20.7 | 34.5 | 48.3 | |
| $\mu$ | 2.83 | 2.86 | 2.87 | 2.90 | 2.94 | 2.96 | 2.98 | |
| $\chi$ | 2.65 | 2.67 | 2.68 | 2.71 | 2.75 | 2.76 | 2.78 | |
| [min–max] | 2.13–4.10 | 2.14–4.11 | 2.15–4.12 | 2.16–4.16 | 2.17–4.23 | 2.17–4.29 | 2.18–4.33 | |

$\lambda$—thermal conductivity, *T*—temperature, *P*—pressure, $\mu$—arithmetic mean, $\sigma$—population standard deviation, $\chi$—median, min and max—minimum and maximum values, respectively.

## 4.2. 2D Subsurface Temperature Distribution

Sensitivity analyses were carried out to assess the influence of GSTH and conditions of the thermophysical properties (dry and water-saturated state) on the subsurface temperature distribution. The statistical distribution of the thermophysical properties was taken into account to run these simulations. Moreover, the effect of pressure and temperature on thermal conductivity was implemented in the models. A deterministic approach was followed, and the minimum subsurface temperature was obtained by combining the maximum value evaluated for thermal conductivity with the maximum value of volumetric heat capacity and radiogenic heat production. The maximum temperature was obtained by using the inverse combination.

### 4.2.1. Influence of Model Mesh

A mesh-dependency study was carried out to guarantee the reliability of the results. The free-triangular mesh was gradually refined until a constant temperature at a given point $(x, z)$ in the model was obtained. This study started with an extremely coarse mesh (22 elements) until a constant temperature at (17,999, −4999.5) was reached for an extremely fine mesh with 8544 elements. However, a mesh with 13,725 elements was used instead to guarantee the correct distribution of the elements throughout the geometry (Table 6). The maximum and minimum element size was set as 250 and 0.5 m, respectively, with a maximum element growth of 1.1 and resolution of 1 in narrow regions.

**Table 6.** Verification of the mesh independence.

| Number of Elements | $T$ (17,999, −4999.5) (°C) | Relative Difference (%) |
|---|---|---|
| 22 | 98.71 | - |
| 289 | 98.56 | −0.15 |
| 758 | 98.52 | −0.04 |
| 2223 | 98.53 | 0.01 |
| 8544 | 98.55 | 0.02 |
| 9750 | 98.55 | 0 |
| 13,725 | 98.55 | 0 |

### 4.2.2. Influence of GSTH

The following temperature simulations were run considering the samples at dry state. The comparison between dry and water saturation conditions is discussed in the next section. The different scenarios for the Laurentide Ice Sheet basal temperature (Table 1) reveal a minimal influence on the subsurface temperature distribution at the base of the model. The difference between the warm and the cold scenarios is up to 1% (Figure 4; Table 7). In the first kilometers, however, the difference is about 80% for the minimum temperature simulated and 14% for the maximum (Figure 4, Table 7). The climate scenarios reveal no influence on the median temperature simulated (Figure 4, Table 7). The uncertainty of the subsurface temperature due to the heterogeneous character of the lithological units is 78% (Figure 4, Table 7). This corresponds to the difference between the maximum and minimum simulated temperatures.

**Table 7.** Subsurface temperature distribution as a function of the GSTH. The reader is referred to Table 1 for further information on the climate scenarios.

| Depth (km) | $T_{min}$ (°C) | | $T_{median}$ (°C) | | $T_{max}$ (°C) | |
|---|---|---|---|---|---|---|
| | Cold | Warm | Cold | Warm | Cold | Warm |
| 0–1 | 1 | 5 | 13 | 13 | 19 | 22 |
| 1–2 | 10 | 15 | 38 | 38 | 62 | 65 |
| 2–3 | 20 | 24 | 63 | 63 | 105 | 107 |
| 3–4 | 30 | 33 | 88 | 88 | 148 | 150 |
| 4–5 | 39 | 42 | 113 | 113 | 191 | 193 |
| 5–6 | 49 | 52 | 137 | 137 | 234 | 235 |
| 6–7 | 59 | 61 | 162 | 162 | 277 | 278 |
| 7–8 | 68 | 70 | 187 | 187 | 320 | 321 |
| 8–9 | 78 | 79 | 212 | 212 | 363 | 363 |
| 9–10 | 88 | 89 | 237 | 237 | 406 | 406 |

*T*—temperature, min—minimum, max—maximum. Minimum, median, and maximum refer to the values evaluated for the temperature considering varying thermophysical properties in each climate scenario.

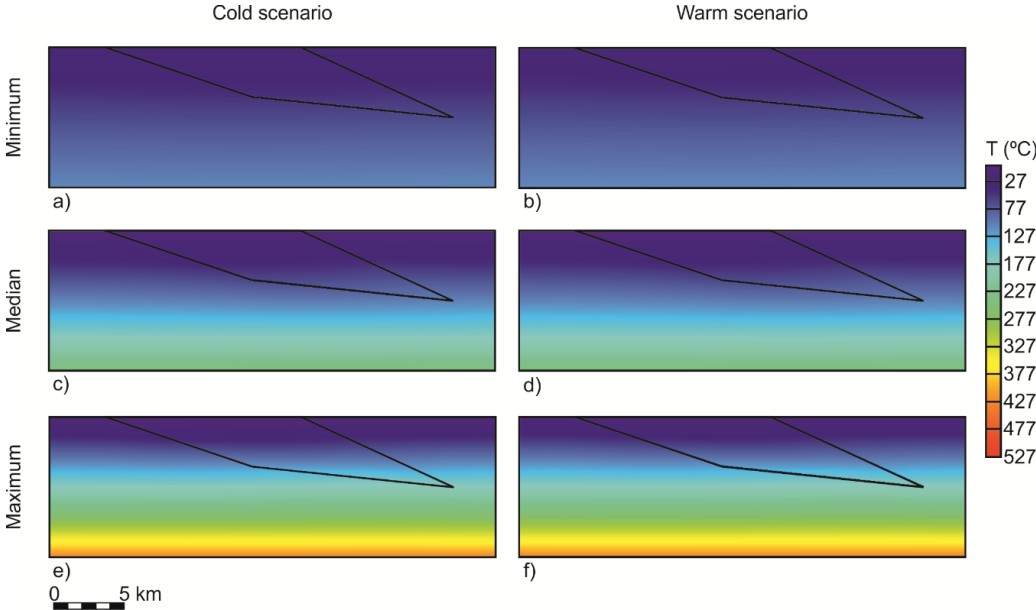

**Figure 4.** 2D subsurface temperature distribution: (**a**) minimum temperature considering the cold scenario; (**b**) minimum temperature considering the warm scenario; (**c**) median temperature considering the cold scenario; (**d**) median temperature considering the warm scenario; (**e**) maximum temperature considering the cold scenario; (**f**) maximum temperature considering the warm scenario. The reader is referred to Table 1 for further information on the climate scenarios. Minimum, median, and maximum refer to the values evaluated for the temperature considering varying thermophysical properties in each climate scenario.

### 4.2.3. Influence of Thermophysical Properties Conditions

The warm climate scenario was chosen to run the following temperature simulations since it revealed the best match between measured and simulated temperature profiles when evaluating the basal heat flux [74]. The water saturation of the thermophysical properties leads to a decrease of the simulated temperature in the minimum and median scenarios (on average, 17–10%, respectively; Figure 5, Table 8). Per contra, the maximum temperature scenario reveals an average increase of 10% (Figure 5, Table 8). The difference between the maximum and minimum temperature is, on average, 78% for the simulations at dry conditions. This difference increases to 84% in the simulations considering water saturation.

**Table 8.** Subsurface temperature distribution as a function of the thermophysical properties conditions.

| Depth (km) | $T_{min}$ (°C) | | $T_{median}$ (°C) | | $T_{max}$ (°C) | |
|---|---|---|---|---|---|---|
| | **Dry** | **Wet** | **Dry** | **Wet** | **Dry** | **Wet** |
| 0–1 | 5 | 5 | 13 | 12 | 22 | 25 |
| 1–2 | 15 | 12 | 38 | 34 | 65 | 72 |
| 2–3 | 24 | 20 | 63 | 57 | 107 | 120 |
| 3–4 | 33 | 28 | 88 | 79 | 150 | 167 |
| 4–5 | 42 | 36 | 113 | 102 | 193 | 215 |
| 5–6 | 52 | 43 | 137 | 124 | 235 | 262 |
| 6–7 | 61 | 51 | 162 | 147 | 278 | 310 |
| 7–8 | 70 | 59 | 187 | 169 | 321 | 358 |
| 8–9 | 79 | 67 | 212 | 191 | 363 | 405 |
| 9–10 | 89 | 74 | 237 | 214 | 406 | 453 |

*T*—temperature, min—minimum, max—maximum. Dry and wet conditions refer to the state at which the thermophysical properties were evaluated. Minimum, median, and maximum refer to the values evaluated for the temperature considering varying thermophysical properties.

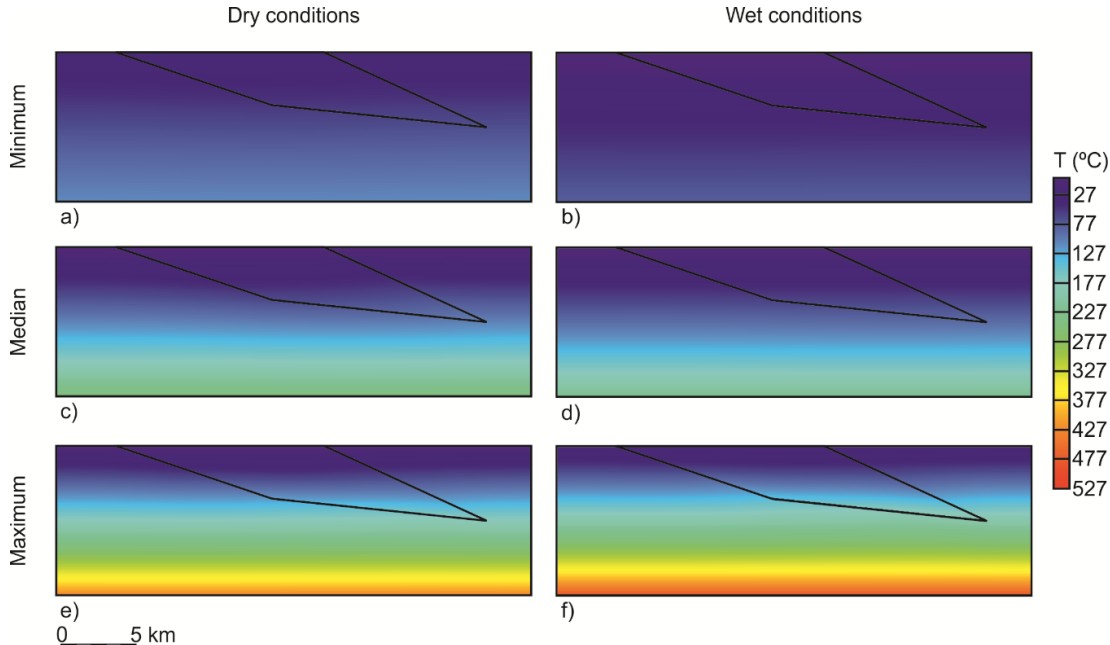

**Figure 5.** 2D subsurface temperature distribution: (**a**) minimum temperature considering the thermophysical properties at dry conditions; (**b**) minimum temperature considering the thermophysical properties at water saturation conditions; (**c**) median temperature considering the thermophysical properties at dry conditions; (**d**) median temperature considering the thermophysical properties at water saturation conditions; (**e**) maximum temperature considering the thermophysical properties at dry conditions; (**f**) maximum temperature considering the thermophysical properties at water saturation conditions. Dry and wet conditions refer to the state at which the thermophysical properties were evaluated. Minimum, median, and maximum refer to the values evaluated for the temperature considering varying thermophysical properties.

### 4.3. Geothermal Energy Source and Potential Heat and Power Output

The geothermal energy source and potential output are examined in two different sections considering the envisioned applications. Results obtained for heat production are first described, followed by electricity generation. Two key questions are answered in both sections:

1. Which geological and technical uncertainties are the most influential input parameters?
2. Can the deep geothermal energy source meet the heat/power demand in Kuujjuaq?

#### 4.3.1. Heat Production

The depth of 1 km was excluded from the following analyses since the 2D subsurface temperature models revealed lower reservoir temperature than the defined reservoir abandonment temperature (30–50 °C; Tables 7 and 8).

Sensitivity analyses were carried out to infer the consistency of the input-output relationship and to compare the relative importance of the input parameters, and thus answering the first aforementioned key question. The Spearman correlation coefficient was evaluated to obtain a qualitative measure of the effect of the uncertain parameters in the potential heat output. A strong positive or negative correlation (i.e., high correlation coefficient) indicates high influence of the input parameter in the output. Per contra, a weak correlation (i.e., low correlation coefficient) suggests a minor influence. The results reveal that volumetric heat capacity and GPP factor have a very weak correlation with the potential heat output, regardless the depth, GSTH, and conditions of the thermophysical properties. The obtained Spearman coefficients range between −5% and 1% for the GPP factor and −1% and 11% for the volumetric heat capacity. The project lifetime and reservoir abandonment temperature are

weakly to moderately correlated with the potential heat output. The former has correlation coefficients ranging between −1% at 2 km depth and −27% at 10 km depth. The correlation coefficients of the latter vary from −44% at 2 km depth to −4% at 10 km depth. The recovery factor and reservoir temperature reveal a moderate to very strong correlation with the potential heat output. The reservoir temperature has correlation coefficients varying between 82% and 48% as a function of depth, while for the recovery factor the coefficients increase with depth from 42% to 80%.

Therefore, due to their low correlation coefficients, any change in the GPP factor and volumetric heat capacity will have a minimal influence on the potential heat output (Figure 6). At 2 km depth, the potential heat output is sensitive to the reservoir abandonment temperature, but this variable loses importance as a function of depth (Figure 6). The significance of the project lifetime increases with depth (Figure 6). Nonetheless, reservoir temperature and recovery factor are clearly the most influential input parameters, regardless of the depth, GSTH, and conditions of the thermophysical properties (Figure 6). The results indicate a switch of rank between reservoir temperature and recovery factor (Figure 6) with the increase of the minimum reservoir temperature when reaching values above the minimum reservoir abandonment temperature (30 °C).

Moreover, the results highlight that decreasing the reservoir abandonment temperature and the project lifetime and increasing the recovery factor lead to an increase in the potential heat output.

The probabilistic approach together with the following assumptions helps to answer the second key question: "Can the deep geothermal energy source meet the heating demand in Kuujjuaq?":

- The average annual heating need is approximately 71 MWh per residential dwelling in Kuujjuaq [31]
- The total number of dwellings is 518 [77]

Thus, this corresponds to an average annual heat consumption of about 37,000 MWh (or, 37 GWh). This value was used as the threshold to assess the probability of the geothermal energy source to meet the community's estimated heating demand (Figure 7). It is convenient to highlight, however, that this approach neglected peak loads as an auxiliary system is more likely to be used to supply heat during peak conditions.

The probability of meeting the estimated heating demand is higher than 98% at a depth of 4 km and below, considering the current geological and technical uncertain parameters along with their distribution span and regardless of the GSTH and conditions of the thermophysical properties (Figure 7d–i). At 2 km depth, the probability of meeting the heating demand ranges from 24.8%, for the cold GSTH, to 33.0%, for the water saturation scenario (Figure 7a). The geothermal energy source at 2 km depth will fulfill the community's needs only if the reservoir temperature is above its 65th–70th percentile and the reservoir abandonment temperature is decreased to values below its 15th–30th percentile (Figure 8). Although the probability of meeting the heating demand at 3 km depth is 80.5% to 83.7% (Figure 7b), the reservoir temperature is required to be higher than its 15th percentile and the recovery factor cannot be lower than the minimum value defined (2%; Figure 8). At 4 km and below, the heating needs are met if the uncertain parameters (mainly reservoir temperature and recovery factor) are within the defined distribution spans (Figure 8).

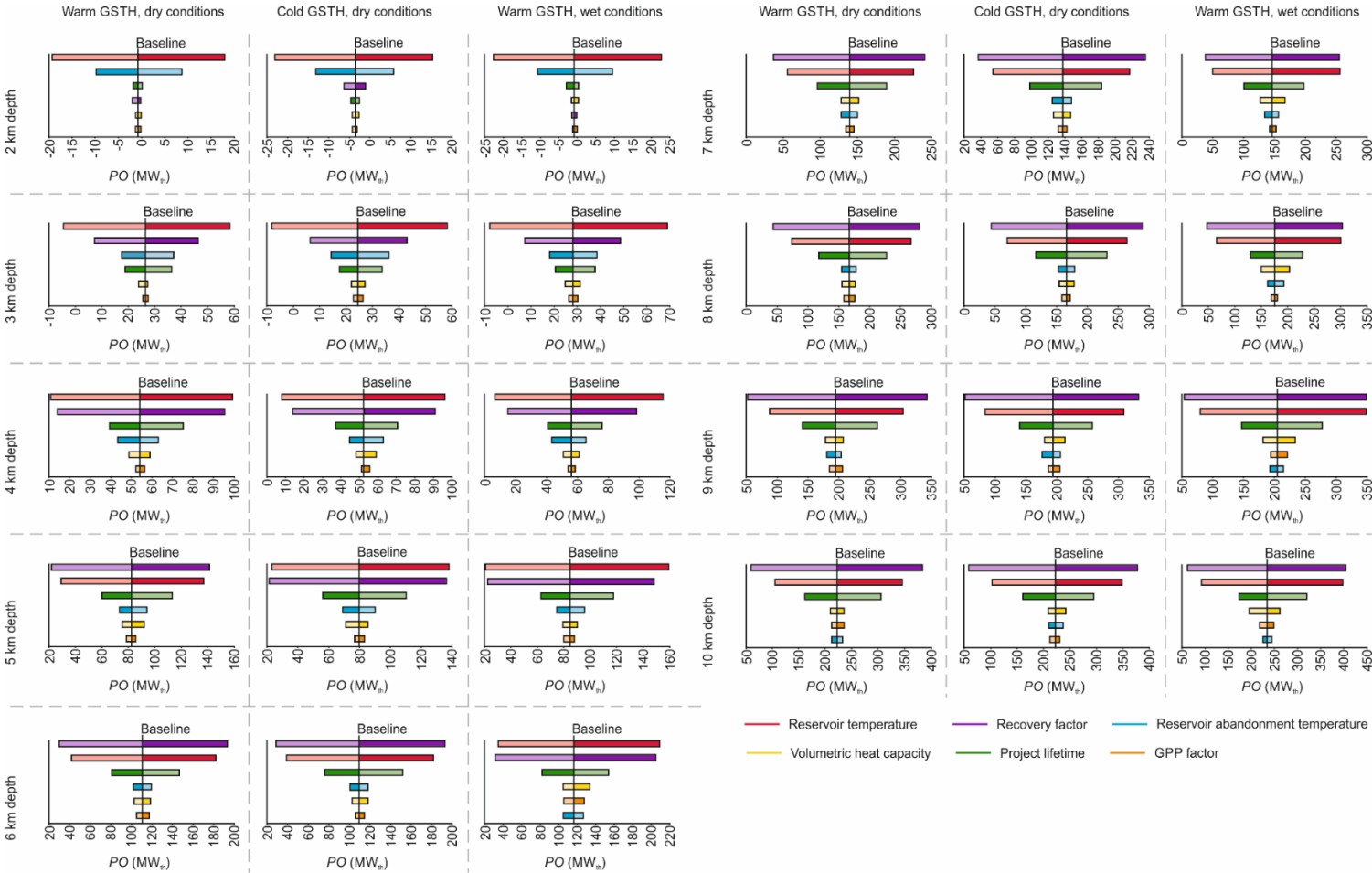

**Figure 6.** Input parameters ranked according to their influence on the geothermal energy source and potential heat output. The reader is referred to Table 1 for further information on the climate scenarios. Dry and wet conditions refer to the state at which the thermophysical properties were evaluated. Baseline—overall simulated mean value; solid color—positive impact on the output; transparency—negative impact on the output. The reader is referred to the online version of the manuscript for further information on the colors.

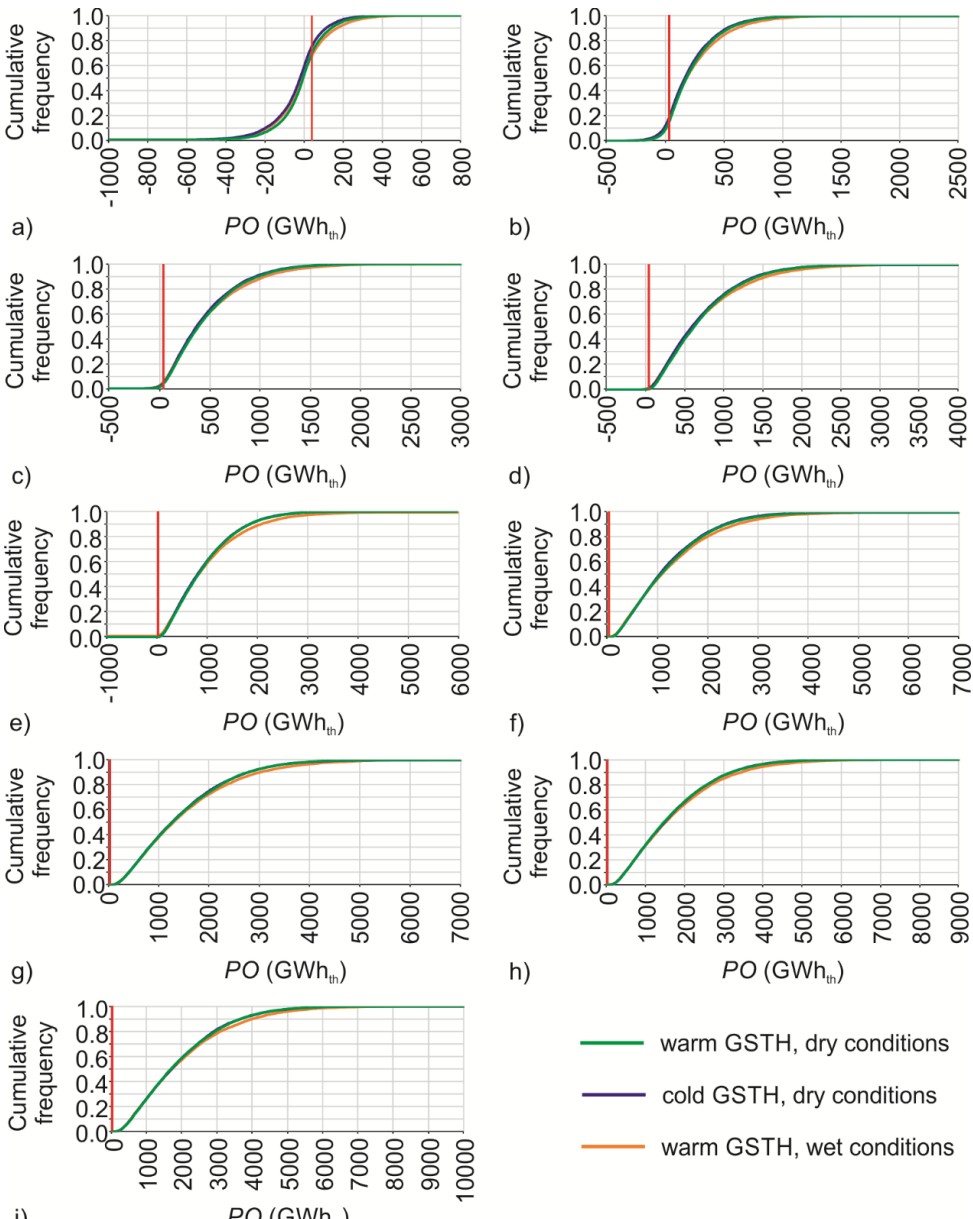

**Figure 7.** Annual geothermal heat output potential and probability of meeting the community's annual average heating demand: (**a**) 2 km depth; (**b**) 3 km depth; (**c**) 4 km depth; (**d**) 5 km depth; (**e**) 6 km depth; (**f**) 7 km depth; (**g**) 8 km depth; (**h**) 9 km depth and (**i**) 10 km depth. Red line—community's estimated heating demand (see text for further details). The reader is referred to Table 1 for further information on the climate scenarios. Dry and wet conditions refer to the state at which the thermophysical properties were evaluated. The reader is referred to the online version of the manuscript for further information on the colors.

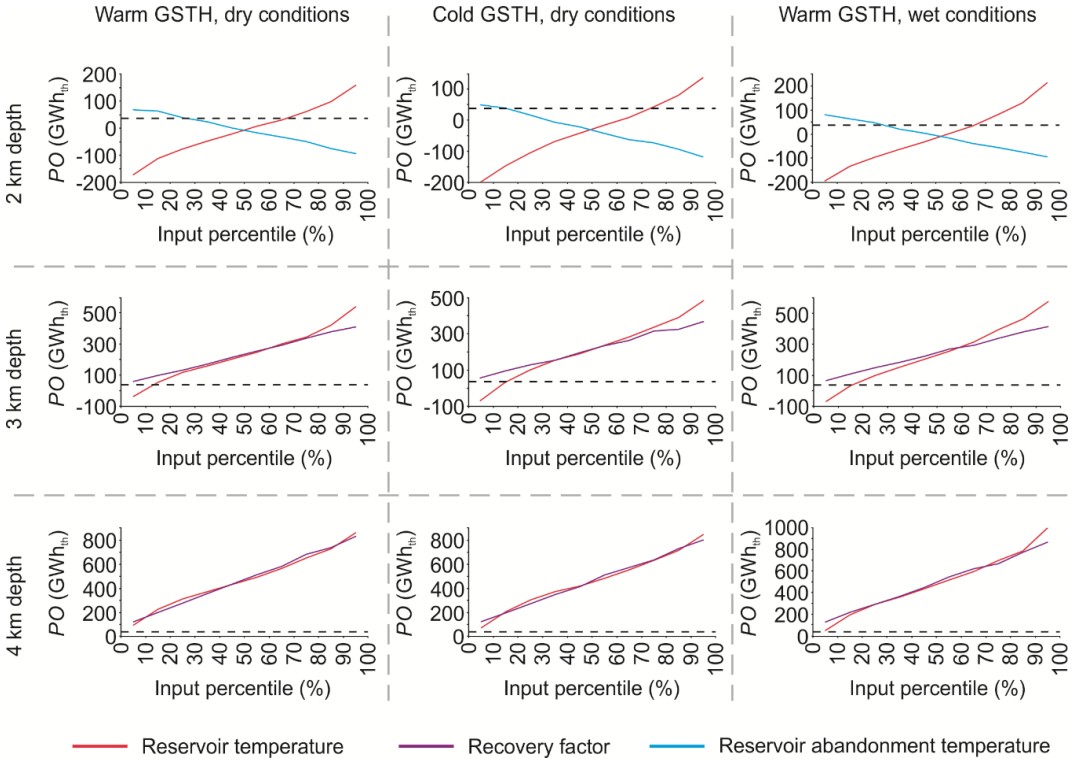

**Figure 8.** Annual geothermal heat output potential as a function of the uncertain parameters' percentile. Dashed line—community's estimated heating demand (see text for further details). The reader is referred to Table 1 for further information on the climate scenarios. Dry and wet conditions refer to the state at which the thermophysical properties were evaluated. The reader is referred to the online version of the manuscript for further information on the colors.

### 4.3.2. Electricity Generation

The following analysis was made at a depth of 5 km and below since the 2D subsurface temperature models revealed lower reservoir temperature than the reservoir abandonment temperature defined for electricity generation (120–140 °C; Tables 7 and 8). Moreover, although the maximum temperature simulated at 4 km is higher than 120 °C (Tables 7 and 8), the potential power output predominantly falls within the negative values leading to biased results.

Similarly to heat production, the Spearman correlation coefficient was also inferred in this section to qualitatively evaluate the input-output relationship. Higher positive/negative correlation coefficient implies more consistency in the relationship than lower coefficient. The relative importance of the input parameters was likewise illustrated with tornado charts.

The results reveal that volumetric heat capacity and GPP factor have a very weak correlation, with the potential power output, regardless the depth, GSTH, and conditions of the thermophysical properties. The correlation coefficients vary within −2% and 2% for the GPP factor and −2% and 5% for the volumetric heat capacity. Hence, the influence of these uncertainties is minimal (Figure 9). The project lifetime has a weak negative correlation with the potential power output ranging between 8% and −16%. The significance of this parameter increases as a function of depth (Figure 9). The reservoir abandonment temperature is negatively weakly correlated to the potential power output, with coefficients varying between −20% and −5%, losing importance as a function of depth (Figure 9). The recovery factor is moderately correlated with the potential power output. The Spearman coefficient ranges between −26% and 48%. This uncertain parameter becomes more influential as a function of depth (Figure 9). Lastly, the reservoir temperature has a moderate to very strong correlation with the potential power output, with coefficients ranging within 37% and 94% and is clearly the most influential parameter for the electricity generation potential (Figure 9).

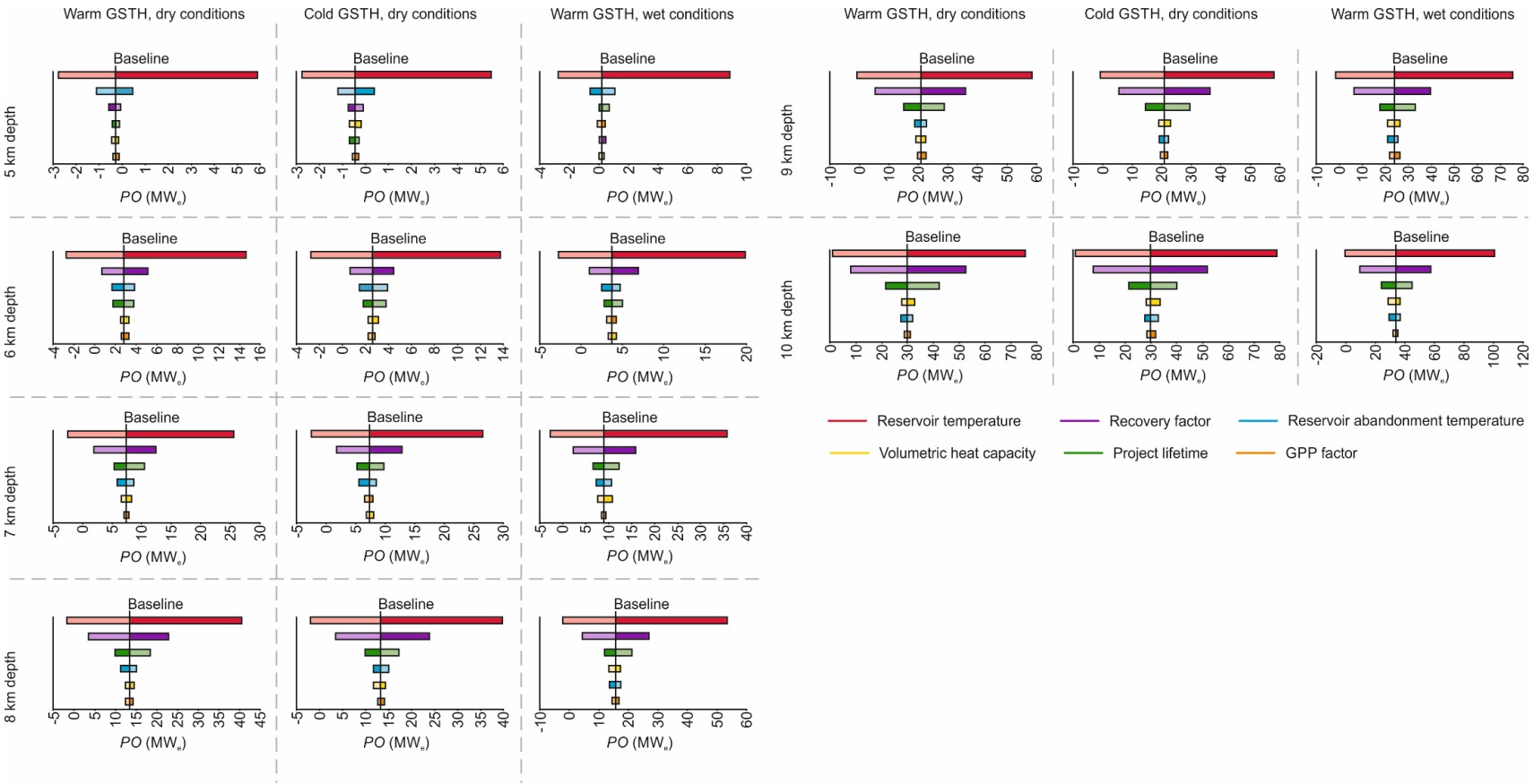

**Figure 9.** Input parameters ranked according to their influence on the geothermal energy source and potential power output. The reader is referred to Table 1 for further information on the climate scenarios. Dry and wet conditions refer to the state at which the thermophysical properties were evaluated. Baseline—overall simulated mean value; solid color—positive impact on the output; transparency—negative impact on the output. The reader is referred to the online version of the manuscript for further information on the colors.

Moreover, the results highlight that decreasing the reservoir abandonment temperature and the project lifetime and increasing the recovery factor leads to an increase in the potential power output.

Combining the probabilistic approach with the following assumptions enables to answer the question: "Can the deep geothermal energy source meet the power demand in Kuujjuaq?":

- In 2011, the electricity consumption in Kuujjuaq was 15,100 MWh [80]
- The population in the community of Kuujjuaq increased about 14% from 2011 to 2016 [77]

Hence, assuming the same growth rate in terms of electricity needs, this corresponds to the current annual consumption of approximately 18,900 MWh (or, 18.9 GWh). This value was used as the threshold to assess the probability of the geothermal energy source to meet the community's estimated power needs (Figure 10). It is important to highlight that the electricity produced and consumed in the community of Kuujjuaq is only for lighting, electrical household appliances, and other electrical devices in service buildings [81]. The space heating and domestic hot water are provided by oil furnaces [81].

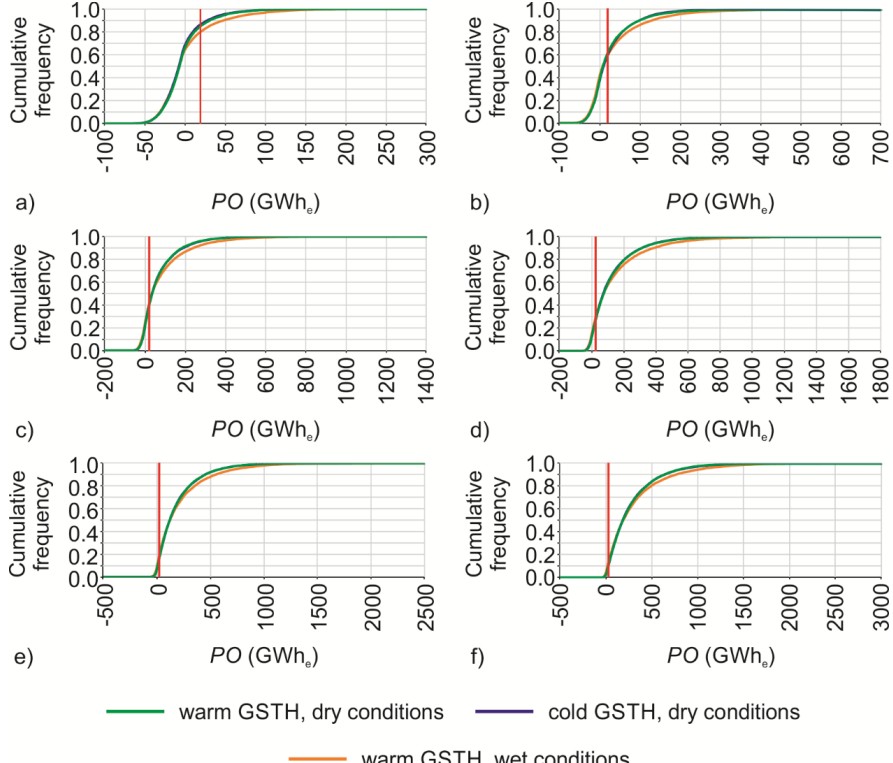

**Figure 10.** Annual geothermal power output potential and probability of meeting the community's electricity demand: (**a**) 5 km depth; (**b**) 6 km depth; (**c**) 7 km depth; (**d**) 8 km depth; (**e**) 9 km depth and (**f**) 10 km depth. Red line—community's estimated electricity demand (see text for further details). The reader is referred to Table 1 for further information on climate scenarios. Dry and wet conditions refer to the state at which the thermophysical properties were evaluated. The reader is referred to the online version of the manuscript for further information on the colors.

The probability of meeting the power demand ranges from 13.6% to 20.2% at 5 km depth (Figure 10) and is between 38.4% and 40.8% at 6 km (Figure 10), considering the current geological and technical uncertain parameters and their distribution span. The highest percentage was obtained for the water saturation conditions while the lowest percentage is associated to the cold GSTH. At depths of 7 km and below, the probability of meeting the power demand is higher than 50% but lower than 100% (Figure 10). The probability is 88% to 91.2% at 10 km, depending on the GSTH and the conditions of the thermophysical properties (Figure 10). The lowest value was obtained for the water saturation state while the highest value for the dry conditions.

Moreover, a detailed analysis was carried out indicating that at 5 km depth, the power demand will be met if the reservoir temperature is higher than its 80th percentile, regardless of the GSTH and the conditions of the thermophysical properties (Figure 11). At 6 km, the reservoir temperature must be higher than its 55th percentile, the recovery factor higher than its 40th percentile, and the reservoir abandonment temperature and the project lifetime lower than their 80th percentile (Figure 11). At 7 km, the demand is met if the reservoir temperature is above its 35th percentile (Figure 11). At 8 km, the reservoir temperature needs to be higher than its 20th percentile and at 9 and 10 km, above its 10th percentile (Figure 11). At 7 km and below, the recovery factor is required to be higher than the minimum defined value (2%; Figure 11).

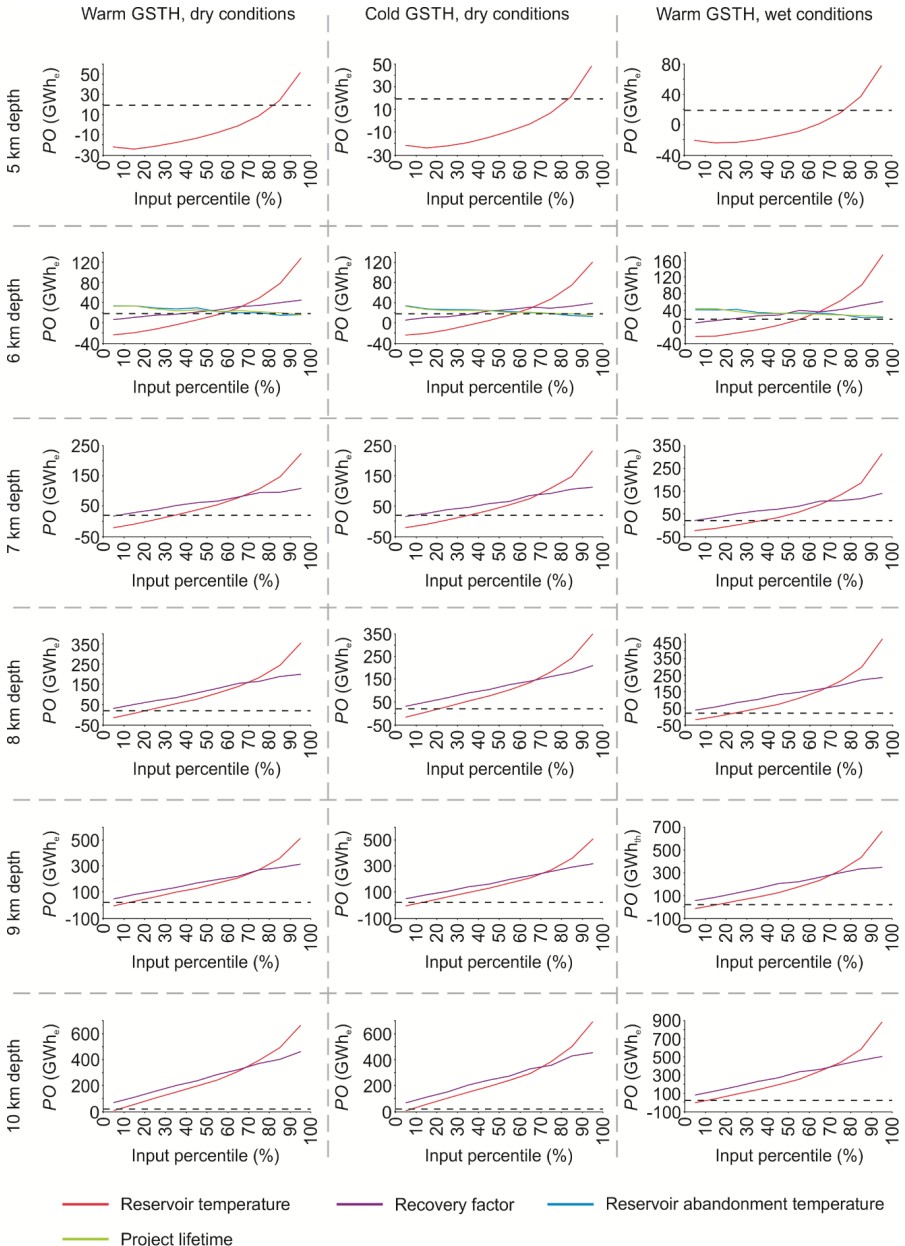

**Figure 11.** Annual geothermal power output potential as a function of the uncertain parameters' percentile. Dashed line—community's estimated power demand (see text for further details). The reader is referred to Table 1 for further information on the climate scenarios. Dry and wet conditions refer to the state at which the thermophysical properties were evaluated. The reader is referred to the online version of the manuscript for further information on the colors.

Cogeneration of heat and electricity, or combined heat and power (CHP), can potentially improve the efficiency of the geothermal energy source. The waste heat from the conversion process can be used for heat supply instead of being discharged to the environment [115–118]. Thus, an analysis was carried out for a depth of 5 km and below to assess if CHP can be an alternative to supply both heat and power.

The results reveal that about 5 to 14 $MW_{th}$ are rejected per each $MW_e$ produced when considering geothermal energy sources in Kuujjuaq (Table 9). The waste heat decreases as a function of depth as the cycle thermal efficiency increases as a function of the reservoir temperature. Although 50% to 60% of this waste heat can only be used for other applications [118] as the remaining is utilized by parasitic equipment requirements [116], the annual heat output potential associated to CHP is significant (Table 9). However, the probability of fulfilling the community's annual heating demand is lower than 90% at 5 km depth, decreasing to less than 12% at 10 km depth (Figure 12).

**Table 9.** Waste heat produced per unit electric capacity ($MW_{th}/MW_e$) and combined heat and power (CHP) heat output potential considering 50% of the waste heat is recovered.

| Depth (km) | | Waste Heat ($MW_{th}$) | | | CHP ($GWh_{th}$) | | |
|---|---|---|---|---|---|---|---|
| | | Warm GSTH, Dry | Cold GSTH, Dry | Warm GSTH, Wet | Warm GSTH, Dry | Cold GSTH, Dry | Warm GSTH, Wet |
| 5 | $\mu$ | 13 | 14 | 14 | 58 | 62 | 62 |
| | $\sigma$ | 6 | 7 | 8 | 26 | 32 | 37 |
| | [min–max] | 6–51 | 6–72 | 6–93 | 28–222 | 28–317 | 25–406 |
| 6 | $\mu$ | 10 | 11 | 11 | 46 | 47 | 48 |
| | $\sigma$ | 4 | 5 | 6 | 19 | 21 | 25 |
| | [min–max] | 5–39 | 5–42 | 5–58 | 22–169 | 23–185 | 20–254 |
| 7 | $\mu$ | 8 | 9 | 9 | 37 | 38 | 38 |
| | $\sigma$ | 3 | 4 | 4 | 15 | 16 | 19 |
| | [min–max] | 4–29 | 4–31 | 4–39 | 19–126 | 19–134 | 17–172 |
| 8 | $\mu$ | 7 | 7 | 7 | 31 | 32 | 32 |
| | $\sigma$ | 3 | 3 | 3 | 12 | 13 | 15 |
| | [min–max] | 4–23 | 4–24 | 3–30 | 16–102 | 16–106 | 14–132 |
| 9 | $\mu$ | 6 | 6 | 6 | 27 | 27 | 28 |
| | $\sigma$ | 2 | 2 | 3 | 10 | 10 | 13 |
| | [min–max] | 3–19 | 3–20 | 3–25 | 14–85 | 14–87 | 12–108 |
| 10 | $\mu$ | 5 | 5 | 6 | 24 | 24 | 24 |
| | $\sigma$ | 2 | 2 | 2 | 9 | 9 | 11 |
| | [min–max] | 3–17 | 3–16 | 3–21 | 12–72 | 12–72 | 11–91 |

$\mu$—arithmetic mean, $\sigma$—standard deviation, min—minimum, max—maximum. The reader is referred to Table 1 for further information on the climate scenarios. Dry and wet conditions refer to the state at which the thermophysical properties were evaluated.

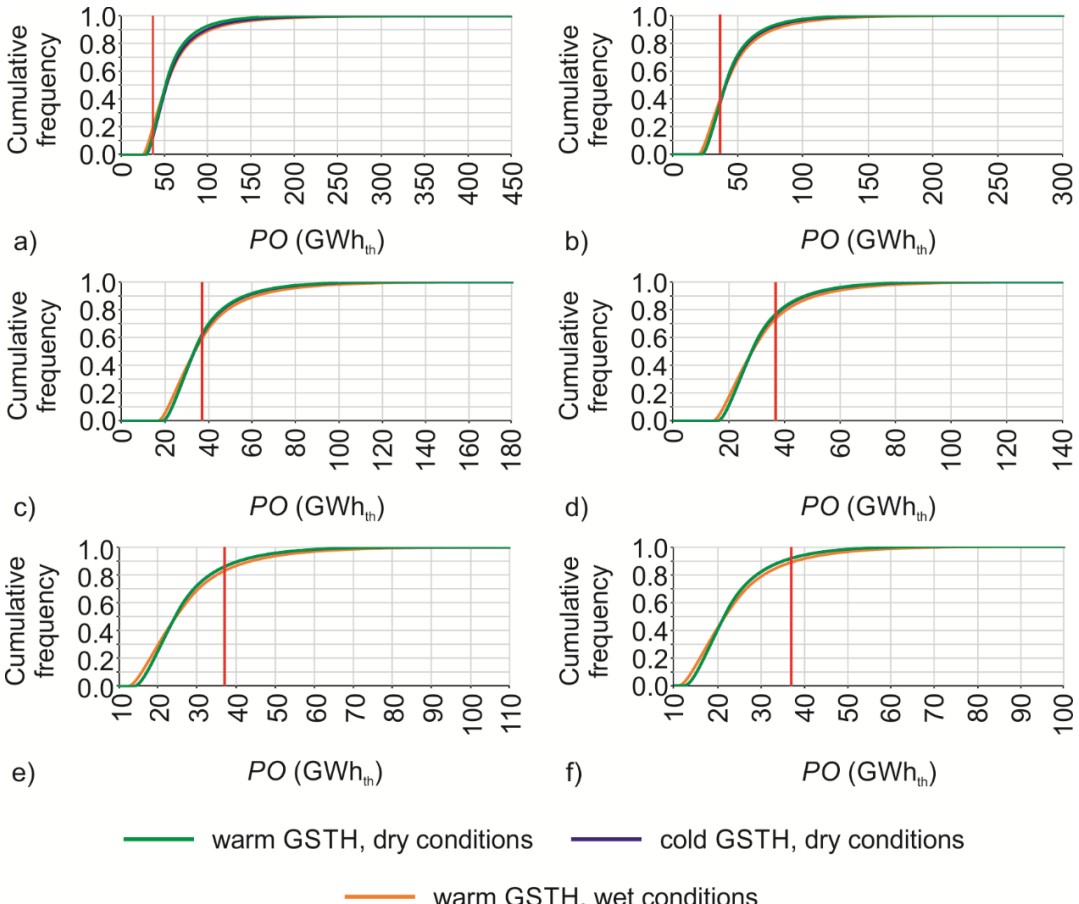

**Figure 12.** Annual CHP heat output potential and probability of meeting the community's heating demand: (**a**) 5 km depth; (**b**) 6 km depth; (**c**) 7 km depth; (**d**) 8 km depth; (**e**) 9 km depth; and (**f**) 10 km depth. Red line—community's estimated heating demand (see text for further details). The reader is referred to Table 1 for further information on the climate scenarios. Dry and wet conditions refer to the state at which the thermophysical properties were evaluated. The reader is referred to the online version of the manuscript for further information on the colors.

## 5. Discussion

The exclusive reliance on diesel for electricity, space heating, and domestic hot water is a reality in Canada, not only in Nunavik but also in the remaining 225 communities spread over Northwest Territories, Nunavut, Yukon, and in the northern part of the provinces of British Columbia, Manitoba, Ontario, and Newfoundland and Labrador [3]. These Arctic and subarctic communities are at the front line of climate change and can play a leading role in providing energy solutions to this national and global challenge. The investment in the geothermal energy sector, for instance, can support local and sustainable economic development by creating business opportunities, improve energy security, ensure price stability, and help reducing greenhouse gas emissions [119]. However, there are still significant challenges in remote northern regions that need to be overcome, for example, at the level of the early stage of geothermal research and development. As mentioned in this study, important data gaps exist in remote areas. Deep boreholes suitable for geothermal energy source assessment are limited to areas of interest for oil and gas and mining exploration [2,38], and thus often located away from the communities where the targeted energy customers are. Therefore, there is a growing need to adapt methodologies and define guidelines in the remote northern regions using outcrops treated as subsurface analogs and shallow data for a first-order assessment of the deep geothermal energy source. This is paramount to stimulate interest to finance further developments and help to advance the

stage of exploration, mainly in the Canadian Shield. This geological province has been assumed less favorable for geothermal development when compared to the western sedimentary basins [2,120–122]. However, these previous geothermal energy source assessments have been based on the extrapolation of limited and uneven distributed heat flux data. Thus, this first-order community-scale research targeting deep geothermal energy source based on surface geological information is an important contribution to help advance the stage of geothermal research and development in remote northern regions. The work has been undertaken in the community of Kuujjuaq (Nunavik, Canada) as a case study to define guidelines that can be further used in the remaining Canadian remote northern communities.

In the following subsections are discussed the weaknesses and strengths of this work, highlighting not only additional envisioned work and future directions but also data limitations that require further geothermal exploration developments for a more accurate evaluation of the deep geothermal energy source. Moreover, the contribution of this work is discussed in the context of the energy transition of remote northern communities located in unconventional geological settings with respect to geothermal energy, and the advantages of deep geothermal energy are compared with other renewable energy sources.

## 5.1. Thermophysical Properties

In the lack of borehole cores, the thermophysical properties of surficial rocks treated as subsurface analogs are essential for a first-order characterization of the deep geothermal energy source [75] and references therein. Therefore, these properties were evaluated, and the results obtained are in the range of values mentioned in literature for similar lithologies of Canadian Shield rocks (e.g., [38,123–125]). Moreover, the values are in accordance with the mineralogical composition and geochemistry of the rock samples (see Miranda et al. [75] for further details). The obtained decrease of thermal conductivity as a function of temperature (Tables 4 and 5) and its increase as a function of pressure (Tables 4 and 5) and water saturation (Table 3) also agree with other experimental studies (e.g., [126–128]).

## 5.2. 2D Subsurface Temperature Distribution

The subsurface temperature distribution was simulated numerically in COMSOL Multiphysics® with the FEM solving the 2D transient heat conduction equation. The geometry of the conceptual model (Figure 3) is based on a schematic regional cross-section of the Ungava Bay area proposed by Simard et al. [87] and thus simplified. Further geothermal exploration developments, such as drilling of exploratory wells and geophysical campaigns (e.g., borehole logging, magnetotellurics; [129]) will be needed to obtain a more complex subsurface geological model.

Nevertheless, this simplified geological conceptual model is helpful for this first-order assessment of the deep geothermal energy source. Moreover, by imposing a time-varying upper boundary condition, the model is capable to reproduce the variations in the surface temperature caused by the several climate events that have been occurring since late Pleistocene (Table 1; [90–106]). These climate disturbances propagate downwards by thermal diffusion, disrupting the steady-state geothermal gradient and affecting not only the present-day heat flux [89] and references therein but the subsurface temperature as well (as supported by this work; Table 7). The amplitude of the climate events exponentially decay with depth and their signal attenuates [55]. Therefore, at a depth below 5 km, the disturbance caused by the paleoclimate is expected to be in phase with the subsurface temperature, as demonstrated by the analytical solution to correct temperature profiles for the paleoclimate effects [130]. The two GSTH scenarios simulated to address the uncertainty associated with the Laurentide Ice Sheet basal temperature (cold and warm; Table 1) revealed a difference between scenarios up to 80% at the surface, progressively decreasing to 1% at the base of the model (Figure 4, Table 7). Therefore, agreeing with the climate signal attenuation with depth. Moreover, the model also reveals that the effects of the climate events are stronger than the lateral variation of the thermophysical properties. The temperature distribution is laterally uniform throughout the model.

The existent subsurface conditions (dry or water saturation) is an unknown that cannot be readily observed at depth. Therefore, two possible scenarios were simulated and compared to deal with this uncertainty. The difference between them is −20 to 10% (Figure 5, Table 8), agreeing with Harlé et al. [76] and highlighting that both scenarios shall be considered when evaluating the deep geothermal energy source.

Beyond the uncertainty imposed by the GSTH and subsurface conditions, the aleatory variability associated with the statistical distribution of the bedrock thermophysical properties was also considered. The three-point estimation technique was used to infer the variability of both the heat flux [89] and the thermophysical properties and, therefore, to presume a minimum, median, and maximum scenarios for the subsurface temperature. The associated uncertainty ranges from 78% to 84%. Deep temperature profiles will enable to ascertain more accurately the heat flux and, subsequently, the subsurface temperature and more complex geostatistical tools can then be applied for the uncertainty quantification [74,131,132]. Nevertheless, the adopted methodology developed by Marquez et al. [88] and improved by Miranda et al. [89] to infer heat flux from shallow temperature logs (80 m), together with the thermophysical properties of the surficial rock samples, is a valuable tool to carry out a first-order assessment of the deep geothermal potential in remote northern communities.

Finally, the effect of both pressure and temperature on thermal conductivity was implemented in the numerical models. The volumetric heat capacity, however, was assumed independent of temperature and a uniform distribution of the radiogenic heat production as a function of depth was considered. Nevertheless, further developments can be envisioned to account for the effect of temperature on the volumetric heat capacity (e.g., [126]) and the possible exponential decay of heat production with depth [133].

*5.3. Geothermal Energy Source and Potential Heat and Power Output*

In this study, the volume method together with a Monte Carlo-based sensitivity analysis was used to evaluate the deep geothermal energy source. The approach followed considered both current geological and technical uncertainties and evaluated the deep geothermal potential in terms of heat production and electricity and cogeneration at each 1 km depth. Three different scenarios were assumed separately to deal with the uncertainty imposed by the GSTH and the conditions of the thermophysical properties (dry vs. water saturation) on the subsurface temperature. The recovery factor is considered a technical uncertainty in this work rather than a geological one [114] since reservoir active volume and flow rate can be optimized by engineering interventions [35].

A key point in the Monte Carlo method is the correct specification of the distribution functions of the input parameters [74,113], as these determine the output response. A single value was defined for the reservoir volume as the goal of this work was to evaluate the deep geothermal energy source at each 1 km depth and within the land surface area occupied by the community (Figure 2). Reservoir simulation and optimization can help to infer the potential microseismic cloud and thus provide a better constraint for the reservoir volume. The triangular distribution function was chosen for the reservoir temperature in the three studied scenarios since the three-point estimation technique was used to define the worst-case, most likely, and best-case subsurface temperature estimations [113]. The Gaussian (normal) distribution was specified for the volumetric heat capacity based on the results from the laboratory analyses. However, the triangular distribution has been commonly assigned for this thermophysical property (e.g., [70]). Further work can be envisioned to compare the outcomes from both distribution functions and evaluate the associated variability. The technical uncertainties related to reservoir abandonment temperature, recovery factor, and GPP factor were assumed to follow a uniform distribution bounded by common minimum and maximum values mentioned in literature. This choice was based on the assumption that, within the defined limit, each outcome of the random variable has equal probability of occurring. The triangular distribution, however, was specified for the project lifetime, assuming 30 years as the most likely value [35,40,41,49].

The sensitivity of the input parameters in this work was determined through a linear regression-based global sensitivity method, using the Spearman correlation coefficient. Moreover, the input parameters were compared and ranked according to their relative importance. This provides a gross, first-order qualitative assessment of the most influential parameters on the deep geothermal energy source and potential heat and power output. Future directions of this work can be envisioned to quantitatively measure the sensitivity through other linear methods, such as the standardized regression coefficients, or through variance-based methods or regionalized sensitivity analysis [74]. Nevertheless, the qualitative results reveal that reservoir temperature and recovery factor are clearly the most influential parameters, and further geothermal exploration should be focused on decreasing their uncertainty (Figures 6 and 9).

*5.4. Deep Geothermal Energy as a Viable Solution to Reduce Fossil Fuels Dependency of Remote Communities*

The following criteria, based on Glassley [109], can be used to evaluate if geothermal energy is a viable alternative solution to fossil fuels in remote northern communities:

1. Is the deep geothermal energy source sufficiently abundant to meet the local heat and power demand?
2. Is the deep geothermal energy source cost-competitive compared to fossil fuels?
3. Will the deep geothermal energy source help to reduce or eliminate greenhouse gas emissions?

The ultimate goal of this work is obviously related to the first fundamental question. A probabilistic approach, accounting for the current geological and technical uncertainties, was used for this purpose and the results revealed deep geothermal energy as a promising solution for the community of Kuujjuaq, especially for heat production. Moreover, the estimated annual average potential power output at 6 km depth (11 GWh$_e$; Figure 10b) is within the range of values inferred by Majorowicz and Grasby [21] for the northwestern sedimentary basins (10–15 GWh).

In the present work, the threshold value used as the community's heating demand did not consider the heat consumed by service buildings (health clinic, shops, hotels, etc.). Only the total residential dwellings (518; [77]) were accounted. Moreover, these were considered as typical 5-occupant household with an annual average heating need of approximately 71 MWh per dwelling [31]. However, 157 of those are single-family houses with an annual average heating demand of about 22 MWh per dwelling [134]. This reduces the estimated annual average consumption from 37 to 29 GWh. In any case, service buildings were left outside from the estimate such that the threshold value represent a fair starting point to evaluate the risk of meeting the heating demand. Thus, the results reveal that, at 4 km depth and below, heat production from deep geothermal energy source is a low-risk application with more than 98% probability of fulfilling the community's heating demand (Figure 7). Moreover, the results show that heat production can be possible at shallower depths of 2 and 3 km if the reservoir temperature is above its 65th–70th percentile and its 15th percentile, respectively (Figure 8). This can help to narrow estimates of a project's capital costs, mainly associated with well drilling. In Nunavut (Canada), a 4 km deep full-size production well can cost approximately $12 million USD (US dollars), increasing up to $30 million USD for an 8 km deep well [37].

A reservoir abandonment temperature of 120–140 °C was selected for the electricity generation analysis based on the operational binary cycle GPP of Kamchatka Peninsula (Russia; [60]). However, Organic Rankine Cycle using an optimized working fluid has helped to achieve favorable electricity generation from lower geothermal energy source of about 80 °C [61–64,135]. Although the efficiency of the GPP with such low heat source is lower than 10% [35], this can be advantageous for remote northern regions where deep drilling cost is high ($12 million USD for a 4-km-well; [37]). Moreover, decreasing the reservoir abandonment temperature leads to an increase in the potential power output. Furthermore, Organic Rankine Cycle has been capable of generating electricity from even lower temperature sources. For example, power generation from waste energy during the process of liquified natural gas regasification was achieved with a source temperature of 100 to 30 °C

(e.g., [136,137]). Thus, Organic Rankine Cycle can be a promising technology for northern regions. However, compared to heat production, electricity generation produced by deep geothermal energy source is a high to medium risk application. The probability of meeting the power needs is low (0.1% at 3 km depth and 94% at 10 km depth), even if the reservoir abandonment temperature is reduced to 80 °C.

Considering the reservoir abandonment temperature threshold of 120–140 °C, the probability of meeting the community's annual power demand is lower than 95% at 10 km depth, with reservoir temperature and recovery factor playing the major roles in constraining the feasibility of such application (Figure 11). A first-order assessment of CHP viability was also undertaken. The waste heat from the power conversion process has low probability to fulfill the community's total heating demand assuming the 37 GWh threshold (Figure 12). However, if this waste heat is only used to provide space heating to half of the residential dwellings, for instance, then the probabilities increase substantially. For example, at 10 km depth, it increases from less than 12% to more than 67%.

Beyond geothermal energy, other alternative solutions to supplant the northern communities' reliance on fossil fuels have been studied. Yan et al. [134] carried out a multi-criteria decision analysis based on the preference ranking organization method for enrichment evaluation method to evaluate the possibility of replacing the traditional heating oil-based systems in Kuujjuaq by either natural gas, biomass, or gasification of domestic waste. Their analysis took into account environmental considerations, social improvements, and economic feasibility and concluded that biomass (using wood pellets) is the favored solution. Thompson and Duggirala [138] compared the economic and environmental costs of electricity generation by biomass CHP, wind, and solar with the traditional diesel engine for off-grid Canadian communities. Their results revealed biomass CHP as the most competitive renewable energy technology. However, in both studies, geothermal energy (shallow or deep systems) was not considered. Often, biomass resources need to be transported and stored similarly to fossil fuels, which is a disadvantage compared to local geothermal energy exploitation [134]. Moreover, wind and solar are highly dependent on weather conditions [139]. Ground-coupled heat pumps and underground thermal energy storage can be viable alternative heating solutions [30,31,33]. However, the energy taken from the subsurface with such shallow systems is generally no more than 50% of the heat needed by a building, requiring an auxiliary system to cover the remaining load [31,33]. Deep geothermal energy sources are the only local alternatives to provide base load heat and electricity, as indicated by this study. Nevertheless, it can be more economic to use an auxiliary system in conjunction with a GPP to supply heat during peak conditions. Future activities can be planned to follow Yan et al. [134] or Thompson and Duggirala [138] methodology comparing biomass resources with deep geothermal energy sources to evaluate which renewable technology is economically, socially, and environmentally best suited for the Canadian remote northern communities.

## 6. Conclusions

Geothermal energy source assessment is an iterative process and imperative to forecast future energy production. In this work, a first-order evaluation of the geothermal energy source and potential heat and power output in a remote northern community of Canada was undertaken based on shallow data and outcrops treated as subsurface analogs. Monte Carlo-based sensitivity analyses were carried out to deal with the current geological (both epistemic and aleatory variability) and technical uncertainties. The statistical distribution of the thermophysical properties due to their intrinsic heterogeneous character is an aleatory variability type. The subsurface temperature, the conditions of the thermophysical properties (dry and water saturation), and the climate signal during a glacial period are epistemic uncertainties. The reservoir abandonment temperature, recovery factor, project lifetime, and GPP factor are technical uncertainties. The study was focused on the community of Kuujjuaq (Nunavik) to provide an example for the remaining off-grid northern communities relying on an unsustainable energetic framework, where fossil fuels are their main source of energy for electricity and space heating.

Thus, a new and alternative approach to conduct geothermal energy source assessment at the community scale based on surface geological information was presented. The knowledge gained can advance the stage of geothermal exploration to take decision on deep drilling. The uncertainty analysis revealed the parameters that have a major impact on the potential heat and power output and the risk analysis highlighted promising geothermal development despite the outcoming uncertainties. Reservoir temperature and recovery factor are the most influential geological and technical uncertainties on the potential heat and power output. Thus, these parameters need to be accurately assessed. Given the current state of knowledge and the high uncertainty, electricity generation, and hence CHP, is high to medium risk applications with less than 92% probability of fulfilling this community with 2750 inhabitants' needs. Heat production, per contra, is a low-risk application at depths of 4 km and below. The probability of meeting the estimated annual average heating demand of the community of Kuujjuaq is higher than 98%, regardless of the GSTH and the conditions of the thermophysical properties.

The results obtained with this study indicate that, although found at a significant depth of at least more than 4 km, the old Canadian Shield beneath the community of Kuujjuaq can host significant geothermal energy source for space heating applications. This is especially important for remote northern communities like Kuujjuaq since this source of energy appears as the only local alternative that can fulfill their heating needs.

To conclude, it is important to highlight that this analysis was based on shallow data and surficial rock samples treated as subsurface analogs. These are low-cost geothermal exploration tools useful for a first-order assessment of the deep geothermal energy source, as indicated by this study. Nonetheless, the stage of geothermal exploration in remote northern regions needs to advance. Deep exploratory wells are essential and the step missing to accurately infer the deep geothermal energy source and potential heat and power output. Thus, helping remote northern communities to move toward a more sustainable and green energetic future.

**Author Contributions:** Conceptualization, M.M.M., J.R., and C.D.; methodology, M.M.M.; validation, M.M.M.; formal analysis, M.M.M.; investigation, M.M.M.; resources, M.M.M., J.R., and C.D.; data curation, M.M.M.; writing—original draft preparation, M.M.M., J.R., and C.D.; writing—review and editing, M.M.M., J.R., and C.D.; visualization, M.M.M.; supervision, J.R. and C.D.; project administration, J.R.; funding acquisition, J.R. All authors have read and agreed to the published version of the manuscript.

**Funding:** This study was funded by the Institut Nordique du Québec (INQ) through the Chaire de recherche sur le potentiel géothermique du Nord awarded to Jasmin Raymond. The Centre d'études Nordiques (CEN), supported by the Fonds de recherche du Québec—nature et technologies (FRQNT), and the Observatoire Homme Milieu Nunavik (OHMI) are further acknowledged for helping with field campaigns cost and logistics.

**Acknowledgments:** The authors would like to acknowledge Félix-Antoine Comeau, Inès Kanzari, Jean-François Dutil, Sérgio Seco, and Stefane Premont for the support during the analyses of the thermophysical properties. Acknowledges are extended to Erwan Gloaguen for the discussions on global sensitivity analysis and Monte Carlo simulation. The authors are also thankful to the two anonymous reviewers whose recommendations helped to improve the manuscript.

**Conflicts of Interest:** The authors declare no conflict of interest. The funders had no role in the design of the study; in the collection, analyses, or interpretation of data; in the writing of the manuscript; or in the decision to publish the results.

## Notation

| | | |
|---|---|---|
| *A* | Radiogenic heat production | $\text{W m}^{-3}$ |
| *b, d* | Experimental constants | |
| *C* | Radioelements concentration | $\text{mg kg}^{-1}$,% |
| *F* | Factor | % |
| *f* | Function | |
| *H* | Thermal energy | J |
| *P* | Pressure | Pa |
| *PO* | Power output | W |
| *Q* | Heat flux | $\text{W m}^{-2}$ |
| *R* | Recovery factor | % |
| *T* | Temperature | °C |
| *t* | Time | s |
| *V* | Volume | $\text{m}^3$ |
| *x, z* | Spatial variables | m |

**Greek letters**

| | | |
|---|---|---|
| $\eta$ | Thermal efficiency | % |
| $\lambda$ | Thermal conductivity | $\text{W m}^{-1}\,\text{K}^{-1}$ |
| $\mu$ | Arithmetic mean | |
| $\rho$ | Density | $\text{kg m}^{-3}$ |
| $\rho c$ | Volumetric heat capacity | $\text{J m}^{-3}\,\text{K}^{-1}$ |
| $\sigma$ | Population standard deviation | |

**Subscript**

| | |
|---|---|
| 0 | Surface |
| 10 | 10-km-depth |
| 20 | Room temperature |
| amb | Ambient |
| dry | Dry conditions |
| e | Electrical |
| g | Ground |
| GPP | Geothermal power plant |
| K | Potassium |
| max | Maximum |
| min | Minimum |
| res | Reservoir |
| ref | Reference |
| Th | Thorium |
| th | Thermal |
| U | Uranium |
| Wet | Water-saturation conditions |

**Abbreviations**

| | |
|---|---|
| ASTM | American Society for Testing and Materials |
| B.P. | Before present |
| CHP | Combined heat and power |
| EGS | Engineered geothermal systems |
| FEM | Finite element method |
| GPP | Geothermal power plant |
| GSTH | Ground surface temperature history |
| IAEA | International Atomic Energy Agency |
| ICP-MS | Inductively coupled plasma-mass spectrometry |
| MIS | Marine Isotope Stage |
| USD | US dollars |
| USGS | United States Geological Survey |

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
