# Peer review of "Uncertainty and Risk Evaluation of Deep Geothermal Energy Source for Heat Production and Electricity Generation in Remote Northern Regions"

_energies, doi:10.3390/en13164221_

Round 1

Reviewer 1 Report

The paper deals with a first order assessment of the geothermal energy source to trigger interest for further development in northern communities.

The paper is interesting and within the scope of the Journal. It is well written and presented. Despite the high uncertainty of the presented results, the authors discuss in detail the assumptions and their consequences making the reader aware of the goal and the conclusions of the work.

Please just take into account the following suggestions to improve the paper.

Introduction

  1. The study carried out in this paper may be extended to other rural/isolated communities. Therefore, the reviewer suggests revising the introduction to generalize it so that it can be applied to all the geographic regions that face similar problems to those of the Nunavik region (volatility and high cost of the diesel price, energy security, and environmental consequences). All the data related to the area of Nunavik (Lines 33-37 53-66 and 73-88) can be introduced in a specific subsection describing the case study (i.e. as a subsection of Section 2).
  2. Future developments of the study should be moved to the conclusions (Lines 69-70, 146-150).
  3. Please clarify if the assumptions considered of Lines 116-121 are related to the works of the literature or the model proposed in this work. In this last case, they have to be included in the Material and Method section, not in the Introduction.

Material and Method

  1. The results reported in Lines 269-272 and those of the mesh-dependency study should be moved to the Results section.

Results

  1. The numerical values of the quantities described in Subsection 4.1 may be reported in a Table to improve the readability.
  2. Please revise Figure 9, 12 to adjust the axis direction.

Author Response

Dear reviewer,

Thank you for your thorough review of the manuscript. Your recommendations helped to greatly improve its quality. In the following paragraphs I have an explanation point-by-point of how I addressed each recommendation you have made.

R: The paper deals with a first order assessment of the geothermal energy source to trigger interest for further development in northern communities.

The paper is interesting and within the scope of the Journal. It is well written and presented. Despite the high uncertainty of the presented results, the authors discuss in detail the assumptions and their consequences making the reader aware of the goal and the conclusions of the work.

Please just take into account the following suggestions to improve the paper.

Introduction

  1. The study carried out in this paper may be extended to other rural/isolated communities. Therefore, the reviewer suggests revising the introduction to generalize it so that it can be applied to all the geographic regions that face similar problems to those of the Nunavik region (volatility and high cost of the diesel price, energy security, and environmental consequences). All the data related to the area of Nunavik (Lines 33-37 53-66 and 73-88) can be introduced in a specific subsection describing the case study (i.e. as a subsection of Section 2).

A: The approach followed in this work can indeed be extended to other off-grid regions. Therefore, the Introduction was modified to generalize the work to, not only Canadian off-grid communities, but to the around 1 billion people still living off-grid worldwide and with limited access to electricity or relying in an unsustainable energetic framework. Lines 33 – 37, 53-66 and 73-88 were modified and the new more generalized text is in Lines 33-43 and 135-137. Lines 55-56 were modified to present the northern Canadian geothermal exploration data gap in a more generalized way. Moreover, new references were added in the text to highlight the electrification challenges faced by Arctic (e.g., Svalbard, Faeroe Islands, Greenland) and non-Arctic communities in south Asia, Africa, Latin America, Europe, etc., and the energy solutions under study in those areas. Although other sources of renewable energy (solar, wind, hydro, etc.) are preferred, the Arctic and non-Arctic off-grid communities are also looking at geothermal energy as another viable and alternative solution. For example, geothermal energy is under study in Svalbard, Greenland, Faroe Islands, Anticosti Island, Caribbean Islands, Azores, Canary Islands, Madagascar, etc. Thus, the study carried out in this work may be helpful for a first order assessment of the deep geothermal energy source in these other Arctic and non-Arctic regions.

All the previous text related to the area of Nunavik was moved to Section 2, that is now designed “Geographical and geological setting”. Figure 1 was also moved from the Introduction to Section 2. Thus, the new text in Section 2 corresponds to Lines 149-176.

R: 2. Future developments of the study should be moved to the conclusions (Lines 69-70, 146-150).

A: Lines 69-70 and 146-150 are not future developments but work carried out in the present manuscript. The sentences were reformulated to avoid a misinterpretation from the reader’s side. The new Lines are 56-60 and 119-121.

R:  3. Please clarify if the assumptions considered of Lines 116-121 are related to the works of the literature or the model proposed in this work. In this last case, they have to be included in the Material and Method section, not in the Introduction.

A: The Lines 116-121 were an explanation of the transient upper boundary condition and the assumption made for the model presented in the manuscript. These lines were repeated further with more detail in the Material and Methods section (Lines 268-301). Therefore, they were erased from the Introduction.

R: Material and Method

  1. The results reported in Lines 269-272 and those of the mesh-dependency study should be moved to the Results section.

A: The results in Lines 269-272 are not part of the work carried out in the present manuscript but an explanation of how the lower boundary condition was inferred that belongs to another manuscript. Therefore, these lines were modified to avoid confusion and Figure 4 was erased. The new text can be read on Lines 258-267. The mesh-independency study was moved to the Results section in a new subsection entitled: “4.2.1. Influence of model mesh”. The corresponding Lines are: 419-428.

R: Results

  1. The numerical values of the quantities described in Subsection 4.1 may be reported in a Table to improve the readability.

A: Three new tables (3, 4 and 5) were added to subsection 4.1 reporting the numerical values. Consequently, the text in this subsection was modified and Figures 5 and 6 were deleted. The new Lines are 372-409.

R: 6. Please revise Figure 9, 12 to adjust the axis direction.

A: Figures 9 and 12 were revised and the orientation of the page was modified so the figures can stay in a landscape layout. The axes are now in the correct direction. Moreover, due to changes in the main text, Figures 9 and 12 are now Figures 6 and 9.

Reviewer 2 Report

Report of the manuscript „Uncertainty and risk evaluation of deep geothermal energy source for heat production and electricity  generation in remote northern regions” by Mafalda M. Miranda, Jasmin Raymond and Chrystel Dezayes.

I have to say that it is a very good feeling to read such a well-described, deep analysis. The manuscript is well-written; the calculations are valid; everything seems to be OK. But, there is a small, but important problem, which should be tackled by the Authors either, probably in the Discussion section. One of the highest cost for the geothermal energy utilization of the drilling, therefore 5 km, 4 km or something even less deep is a crucial point. The Authors assumed (line 128), that in Arctic climate only heat sources around or above 120-140 Celsius can be utilized to produce electricity. It is a common misconception. For electricity production, the temperature of the heat source as well as the temperature of the heat sink is equally important. In Arctic condition, the heat sink temperature is quite low, which is favourable, for example, in Organic Rankine Cycles. I would recommend for the Authors to cite some papers dealing with electricity production by ORC in LNG regasifying stations. Sometimes they have heat source temperatures only around or below 40 Celsius (usually air or sea-water temperature), but the heat sink temperature can be massively below zero. Although even in Arctic conditions, one cannot reach LNG temperatures, but by choosing a proper working fluid, low-temperature geothermal heat (let's say, something between 40-60 Celsius) can be turned to electricity with sufficient (>10%) efficiency. This relatively low efficiency would be compensated by the low cost of drilling; one can obtain 40-50 Celsius at a depth of 1.5 km, instead of 4-5 km.

Concerning LNG “cold energy” recovery, I would recommend checking Dutta et al, ACS Sustainable Chem. Eng. 2018, 6, 10687−10695 or Yu et al (Energy 167 (2019) 730-739), while for the task to find working fluid for cold ORC-s, Imre et al., Energies 2020, 13, 1519; doi:10.3390/en13061519.

Also, I would recommend for the Authors to cite the recent paper of Shi and Pan (Energies 2019, 12, 732; doi:10.3390/en12040732); they are dealing with the utilization of low-temperature geothermal heat, using mixed working fluid they were able to utilize heat source with 90 Celsius, while the heat sink was at ambient (but not in Arctic) temperature and obtained more than 6 % cycle efficiency, which is also not hight, but due to the high drilling cost, still might be a viable alternative.

I believe that it would be crucial to mention, that it might be possible to reach the goal with less deep wells; therefore I would like to ask the Authors to insert one or two paragraphs about this topic into the discussion section, before the acceptance of the manuscript.

Author Response

Dear reviewer,

Thank you for your thorough review of the manuscript. Your recommendations helped to greatly improve its quality. In the following paragraphs I have an explanation point-by-point of how I addressed each recommendation you have made.

R: Report of the manuscript „Uncertainty and risk evaluation of deep geothermal energy source for heat production and electricity generation in remote northern regions” by Mafalda M. Miranda, Jasmin Raymond and Chrystel Dezayes.

I have to say that it is a very good feeling to read such a well-described, deep analysis. The manuscript is well-written; the calculations are valid; everything seems to be OK. But, there is a small, but important problem, which should be tackled by the Authors either, probably in the Discussion section. One of the highest cost for the geothermal energy utilization of the drilling, therefore 5 km, 4 km or something even less deep is a crucial point. The Authors assumed (line 128), that in Arctic climate only heat sources around or above 120-140 Celsius can be utilized to produce electricity. It is a common misconception. For electricity production, the temperature of the heat source as well as the temperature of the heat sink is equally important. In Arctic condition, the heat sink temperature is quite low, which is favourable, for example, in Organic Rankine Cycles. I would recommend for the Authors to cite some papers dealing with electricity production by ORC in LNG regasifying stations. Sometimes they have heat source temperatures only around or below 40 Celsius (usually air or sea-water temperature), but the heat sink temperature can be massively below zero. Although even in Arctic conditions, one cannot reach LNG temperatures, but by choosing a proper working fluid, low-temperature geothermal heat (let's say, something between 40-60 Celsius) can be turned to electricity with sufficient (>10%) efficiency. This relatively low efficiency would be compensated by the low cost of drilling; one can obtain 40-50 Celsius at a depth of 1.5 km, instead of 4-5 km.

Concerning LNG “cold energy” recovery, I would recommend checking Dutta et al, ACS Sustainable Chem. Eng. 2018, 6, 10687−10695 or Yu et al (Energy 167 (2019) 730-739), while for the task to find working fluid for cold ORC-s, Imre et al., Energies 2020, 13, 1519; doi:10.3390/en13061519.

Also, I would recommend for the Authors to cite the recent paper of Shi and Pan (Energies 2019, 12, 732; doi:10.3390/en12040732); they are dealing with the utilization of low-temperature geothermal heat, using mixed working fluid they were able to utilize heat source with 90 Celsius, while the heat sink was at ambient (but not in Arctic) temperature and obtained more than 6 % cycle efficiency, which is also not hight, but due to the high drilling cost, still might be a viable alternative.

I believe that it would be crucial to mention, that it might be possible to reach the goal with less deep wells; therefore I would like to ask the Authors to insert one or two paragraphs about this topic into the discussion section, before the acceptance of the manuscript.

A: The reservoir abandonment temperature of 120 – 140 °C was selected for the electricity generation analysis based on the operational binary cycle GPP of Kamchatka Peninsula (Russia). Given the similar Arctic climate, this geothermal power plant seemed a good example to follow. However, as the reviewer referred in the comment, ORC with an optimized working fluid can indeed generate electricity from lower temperature geothermal energy sources than the threshold value we assumed, which is indeed beneficial for remote northern communities. By decreasing the reservoir abandonment temperature, we can increase the potential power output, and by using lower temperature geothermal energy sources, we can decrease the drilling depth and cost, which is a key point in remote regions. However, due to the low efficiency, electricity generation in Kuujjuaq, and perhaps in other remote communities with a similar geological context, is a medium to high risk application compared to heat production. Lines 98-99 were added in the Introduction mentioning ORC and optimized working fluid and how it helped to achieve favorable electricity generation from low temperature geothermal resources. Moreover, in Lines 852-866 this topic is discussed.

Round 2

Reviewer 2 Report

The Authors adressed all problematic points in proper manner, therefore the manuscript can be accepted now.